# Restructured membrane contacts rewire organelles for human cytomegalovirus infection

Katelyn C. Cook ®[1], Elene Tsopurashvili[1], Jason M. Needham ®[2], Sunnie R. Thompson[2] & Ileana M. Cristea ®[1] ✉

Membrane contact sites (MCSs) link organelles to coordinate cellular functions across space and time. Although viruses remodel organelles for their replication cycles, MCSs remain largely unexplored during infections. Here, we design a targeted proteomics platform for measuring MCS proteins at all organelles simultaneously and define functional virus-driven MCS alterations by the ancient beta-herpesvirus human cytomegalovirus (HCMV). Integration with super-resolution microscopy and comparisons to herpes simplex virus (HSV-1), Influenza A, and beta-coronavirus HCoV-OC43 infections reveals time-sensitive contact regulation that allows switching anti- to pro-viral organelle functions. We uncover a stabilized mitochondria-ER encapsulation structure (MENC). As HCMV infection progresses, MENCs become the predominant mitochondria-ER contact phenotype and sequentially recruit the tethering partners VAP-B and PTPIP51, supporting virus production. However, premature ER-mitochondria tethering activates STING and interferon response, priming cells against infection. At peroxisomes, ACBD5-mediated ER contacts balance peroxisome proliferation versus membrane expansion, with ACBD5 impacting the titers of each virus tested.

Organelle remodeling is an essential component of all human virus infections and tightly linked to viral pathologies. As obligate intracellular parasites, all viruses rely on the biological processes partitioned within organelles for the progression through their replication cycles and the spread of infections[1,2]. These remodeling events underlie virus entry and trafficking, inhibition of immune signaling, and modulation of metabolism needed for the replication, assembly, and egress of new virions. Likewise, host cells require organelles to detect and combat pathogen invasion. Therefore, the finely tuned regulation of organelle structure and function across space and time is fundamental to virus-host interactions and the outcome of a viral infection.

The infectious cycle of the ancient beta-herpesvirus human cytomegalovirus (HCMV), a prevalent human pathogen that latently infects >70% of the adult population and poses significant burden for immunocompromised and pregnant individuals[3], is a powerful example of infection-driven organelle structure-function modulation. HCMV remodels all major organelles during its four-to-five day (96–120 h) replication cycle (Fig. 1A)[4]. During entry into the host cell and establishment of infection, HCMV modulates the plasma membrane, endosomal sorting machinery, and host immune pathways[3,5–8]. Following the nuclear viral genome replication, the nuclear lamina and envelope are disrupted to facilitate capsid egress[9,10], resulting in a characteristic kidney-bean nuclear morphology surrounding a cytoplasmic viral assembly complex (AC), where viral particles mature, gain envelopes, and are targeted for cellular exit. AC formation requires the restructuring of the Golgi apparatus into a cylindrical structure that is filled with cholesterol-rich endosomes and surrounded by dense endoplasmic reticulum (ER) networks[11–15]. Facilitating these alterations and virus production is an extensive host cell metabolic reprogramming driven by virus-mediated changes in the

[1]Department of Molecular Biology, Princeton University, Princeton, NJ 08544, US. [2]Department of Microbiology, University of Alabama Birmingham, Birmingham, AL 35294, US. ✉e-mail: icristea@princeton.edu

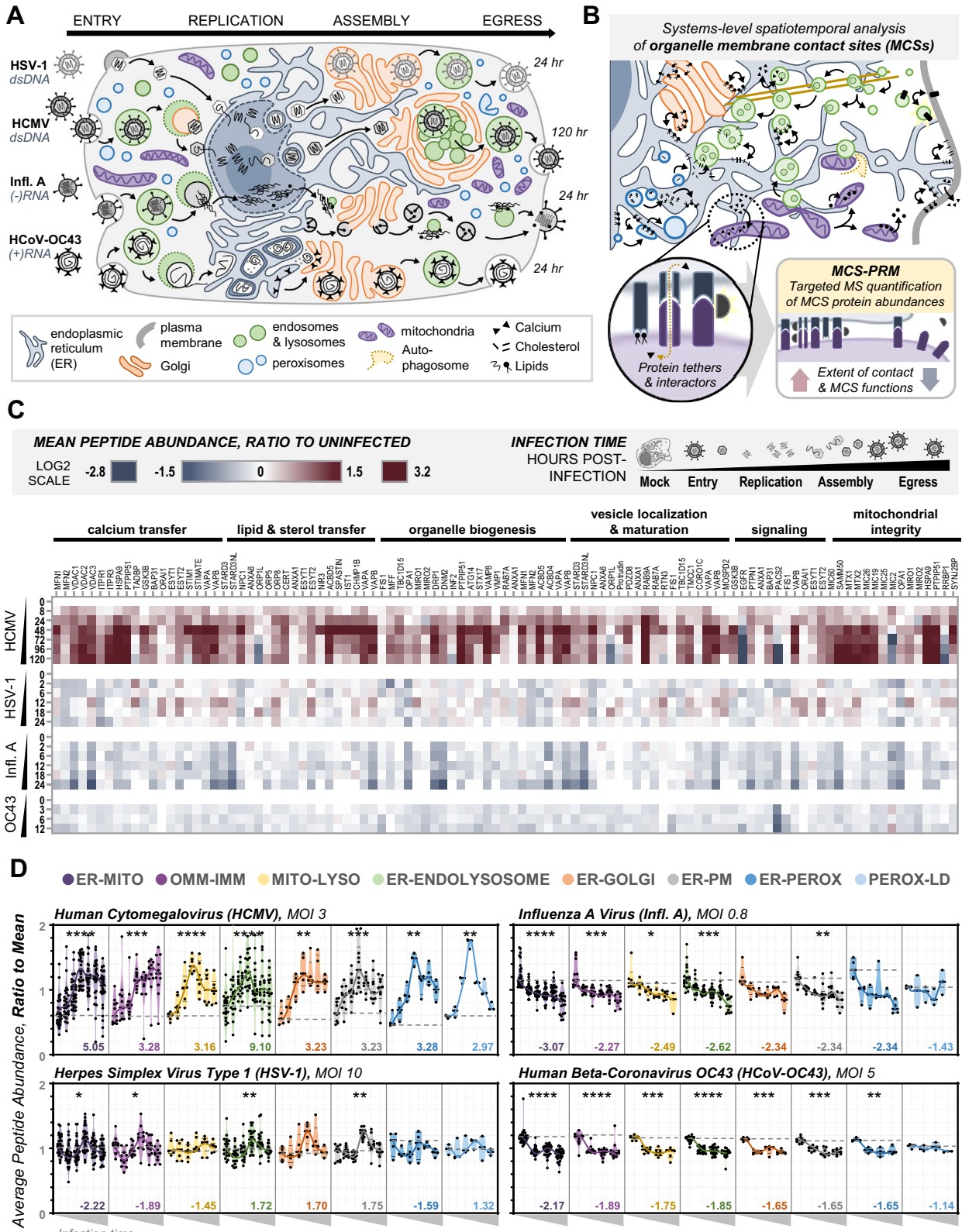

number and morphology of mitochondria and peroxisomes that lead to elevated lipid and ATP production[16–20].

Other viruses alter organelle structure-function relationships to differing extents that reflect their unique replication strategies[2]. For example, the alpha-herpesvirus herpes simplex virus type 1 (HSV-1) increases peroxisome biogenesis, rewires metabolism, and remodels the nuclear envelope and secretory organelles for virion assembly and

trafficking[10,21,22]. However, HSV-1 has a shorter (24-hour) replication cycle, reflected in overall milder changes to organelle structures when compared to HCMV. Similarly, it is well-recognized that rapidly evolving viruses, such as Influenza A (Infl. A) and the beta-coronavirus HCoV-OC43, rely on organelle remodeling for their replication cycles (Fig. 1A). Infl. A infections cause rapid mitochondrial fragmentation–disrupting membrane potential, anti-viral sensing, and apoptotic

**Fig. 1 | HCMV globally modulates organelle contact protein abundances in contrast to infections with HSV-1, Infl. A, and HCoV-OC43. A** Schematic highlighting the organelle remodeling events that underlie the unique infectious cycles of HCMV, HSV-1, Infl. A, and HCoV-OC43, which vary in structure (*left*), size, and replication timescale (*right*). **B** Membrane contact sites, facilitated by protein interactions, enable the direct transfer of biomolecules to coordinately regulate organelle structure and function. MCS-PRM simultaneously quantifies the abundances of MCS protein tethers, regulators, and functional interactors across organelles, identifying alterations to the extent and functions of organelle contacts. **C** MCS-PRM quantification of MCS protein abundances during infections (MOI: multiplicity of infection, i.e., number of viral particles per cell) with HCMV (MOI: 3, $N = 6$), HSV-1 (MOI = 10, $N = 4$), Infl. A (MOI = 0.8, $N = 3$), and HCoV-OC43 (MOI = 5, $N = 3$). Timepoints are indicated at *left*, proteins measured are *top*, grouped by primary function, and heatmap key is *above* ($Log_2$ scale). The rationale for proteins included in MCS-PRM can be found in Supplementary Table 1. **D**. Average peptide abundances (scaled to mean = 1) of MCS proteins grouped by localization (color/order key at *top*) and plotted across infection time (x-axis, grey triangles). Black points are proteins, bold lines connect the median of each timepoint, dotted grey lines represent median abundance in the uninfected state, and numbers indicate the maximum fold-change observed for the given MCS (****$p \le 0.0001$, ***$p \le 0.001$, **$p \le 0.01$, *$p \le 0.05$, N.S. is $p > 0.05$ by two-way ANOVA across all timepoints and a Dunnett's multiple comparisons test to Mock). Specific p-values are as follows, from left to right (ER-Mito to Perox-LD) for each infection: HCMV $p \le 0.0001$, $p = 0.0004$, $p \le 0.0001$, $p \le 0.0001$, $p = 0.0024$, $p = 0.0002$, $p = 0.0026$, $p = 0.0018$; HSV-1 $p = 0.0269$, $p = 0.0373$, N.S., $p = 0.0086$, N.S., $p = 0.0025$, N.S., N.S.; Influenza A $p \le 0.0001$, $p = 0.0008$, $p = 0.0274$, $p = 0.0001$, N.S., $p = 0.0052$, N.S., N.S.; HCoV-OC43 $p \le 0.0001$, $p \le 0.0001$, $p = 0.0004$, $p \le 0.0001$, $p = 0.0003$, $p = 0.0002$, $p = 0.0046$, N.S.

signaling—and later hijack endosomes and plasma membrane, where viruses assemble and egress[23,24]. These subcellular changes contribute to the cytopathic effects and tissue damage induced by Infl. A[25]. Infections with HCoV-OC43 are also known for inflammation and necrosis, yet organelle alterations remain largely uncharacterized beyond ER restructuring into double-membraned vesicles (DMVs), which underlie virus genome replication and capsid assembly[26]. ER DMVs are a hallmark of infections with beta-coronaviruses, including SARS-CoV-2 that also disrupts mitochondria and peroxisomes[27–29]. Therefore, understanding virus-driven subcellular remodeling can identify unique and shared features of virus replication cycles. However, the molecular underpinnings of virus-driven organelle rewiring remain largely unknown.

Many organelle alterations observed during viral infections are reminiscent of processes controlled by membrane contact sites (MCSs) (Fig. 1B). MCSs use proteins—including tethers, regulatory partners, and functional interactors—to closely (10-30 nm) link organelles in dynamic intracellular networks[30,31]. All organelles communicate via MCSs of stable or transient nature, with the ER considered the "master regulator" of MCS biology by directly connecting every other organelle in the cell[32]. MCSs facilitate the recruitment and transfer of biomolecules for the cooperative regulation of cellular homeostasis and response to environmental changes (Supplementary Fig. 1A). Specifically, MCSs control organelle structure and biogenesis, metabolism, vesicular trafficking, apoptosis, and lipid organization, among others (reviewed in[33]; references in Supplementary Table 1). Therefore, it is perhaps surprising that MCSs have not been a focus of human infection studies. For other diseases, such as cancer and neurodegeneration, the discovery of MCSs has already benefitted our understanding of cellular pathologies[34–39]. Yet, beyond a handful of isolated reports[40–45], it remains unclear how MCSs contribute to organelle remodeling during human virus infections, such as the global remodeling events observed during HCMV infection.

A major challenge in capturing the breadth of MCS biology in the context of human disease is a lack of systems-level tools that encompass the spatial, temporal, and multifunctional components of MCSs. Most studies to date, including recent methodology advances, have been geared towards discovering the components of organelle contacts[46–49]. These methods are valuable for the continued identification of MCS proteins, structures, and functions, yet are limited in scope to monitoring individual organelle junctions and single biological conditions. Microscopy advances have begun to tackle the need for increased temporal and spatial breadth[50,51], but are often restricted to immortalized cell cultures and do not provide the necessary protein-specific information to pinpoint changes in MCS functions. To effectively translate MCS biology to human disease, there is a need for high-throughput methods that can simultaneously detect all MCSs with functional specificity and enable their quantification across subcellular space and biological time. Such techniques must be adaptable

to diverse model systems as new contacts and functional MCS components continue to be identified.

Here, we design a high-throughput targeted mass spectrometry (MS) platform uniquely suited for profiling MCSs *en masse* during dynamic biological processes. Our tool provides quantification metrics for functional MCS proteins, including tethers, interactors, and functional partners at all major organelles. We demonstrate the ability of this method to capture the scope of MCS-coordinated regulation of organelle structure and function during infection with the prominent DNA viral pathogen HCMV. We next compare HCMV-induced MCS modulations to changes during infections with other DNA (HSV-1) and RNA (Infl. A, HCoV-OC43) viruses. We find that the spatiotemporal modulation of organelle contacts is a shared feature of these pathogen infections. Integrating our findings with live-cell microscopy and virology assays, we establish that the time-sensitive control of ER-mediated MCSs regulates pro-viral alterations to mitochondria and peroxisomes. HCMV temporally restructures ER-mitochondria contacts to support virus production, facilitating evasion of STING immune signaling and later forming a mitochondria-ER encapsulation structure that we name MENC. ER-peroxisome contacts, mediated by ACBD5, increase throughout HCMV infection for viral control of peroxisome size and numbers. Altering ER contact dynamics with either mitochondria or peroxisomes restricts HCMV production, as well as the replication capacities of HSV-1, Infl. A, and HCoV-OC43.

## Results

### A platform for globally defining dynamic alterations to MCSs
MCSs are formed by protein interactions that bridge organelle membranes. The abundance of MCS-specific proteins (e.g., tethers, functional partners, associated complexes) at an organelle interface—which can be influenced by cellular expression, protein localization, and post-translational modifications[52]—dictates both the extent of contact between organelles and the potential functions of the interaction[31]. Therefore, we sought to develop an assay that can quantify proteins across all MCSs simultaneously, without prior sample perturbation, allowing for direct comparisons of native protein abundances across conditions.

Given their low cellular levels and membrane-associated biochemical properties, many MCS proteins are challenging to quantify via traditional MS methods, such as data-dependent acquisition (DDA) approaches (Supplementary Fig. 1B). We thus turned to a targeted MS approach using parallel reaction monitoring (PRM), which allows for precise quantifications of difficult-to-detect proteins in complex samples[53,54]. PRM selectively monitors unique peptide signatures for proteins of interest under experimentally determined detection parameters. To develop a PRM-based assay for MCSs, we first curated a list of MCS proteins reported in mammalian tissues, focusing on functionally-characterized MCS proteins with defined tethering partners and contact-dependent functions (rationale in Supplementary Table 1). We then

experimentally determined a library of peptides unique to these proteins, defining signature parameters for LC separation, MS detection, and MS/MS fragmentation that enable their accurate measurement within a single sample (Supplementary Fig. 1C-F, Supplementary Data 1). Our MCS-PRM library includes every membrane-bound organelle (and lipid droplets), being able to identify parameters for an estimated 90% of known functional MCS proteins, with 2-5 peptides per protein and endogenous peptide controls for robust quantification. The MCS-PRM assay can be adapted to multiple model systems, as we have done here for human lung fibroblasts (MRC5) and primary human renal proximal tubule epithelial (RPTE) cells, cell cultures used for studying herpesvirus and coronavirus infections[55,56], respectively. Our results showed that MCS-PRM requires little starting material (e.g., <1 well of a 12-well plate or 1.5 μg total cell lysate) and is applicable to *en masse* quantification of MCSs across time, giving us confidence that this assay is applicable to monitoring MCSs during dynamic biological contexts.

## HCMV infection globally increases organelle contact proteins

Having validated our MCS-PRM assay, we next quantified MCS protein abundances throughout infection with HCMV, which we further compared to infections with HSV-1, Infl. A, and HCoV-OC43. We expected MCSs to be altered in accordance with each unique viral replication cycle (Fig. 1A), acting as conserved factors that underlie viral abilities to remodel organelle structures and functions. We infected fibroblasts with HCMV, HSV-1, and Infl. A, and epithelial cells with HCoV-OC43, collecting samples at timepoints corresponding to virus entry, genome replication, virion assembly, and cellular egress. Each virus replicates on a different timescale, reflected by our choice of hours post-infection (hpi) timepoints: 0-120 for HCMV; 0-24 for HSV-1 and Infl. A; and 0-12 for HCoV-OC43. Viral proteins were monitored in parallel to confirm progression through virus replication (Supplementary Fig. 2A, Supplementary Data 2).

We find that HCMV triggers a nearly global increase in MCS protein abundances across infection time, with most proteins increasing 2-to-5-fold (Fig. 1C). In contrast, influenza downregulates many MCS proteins by approximately 2-fold, and HCoV-OC43 and HSV-1 cause overall mild changes. Changes in MCS protein abundances were significant and reproducible (Supplementary Fig. 2B, C). We also compared alterations in MCS protein abundances with overall changes in corresponding organelle proteomes and/or functional classes by analyzing, in parallel, whole-cell proteomes during the progression of each infection (Supplementary Figs. 3-5, Supplementary Data 3). Overall, changes in MCS protein abundances were enriched compared to non-MCS proteins of similar localization and functions, particularly for HCMV infection (Fig. 5). These results suggest that, during HCMV infection, MCS protein abundances are selectively altered over proteins of shared subcellular compartments or functional pathways. Although significant, the changes for the other viruses tested were milder and often targeted to specific organelle localizations or functional pathways (e.g., peroxisome, mitochondria, and plasma membrane) (Supplementary Figs. 3B, C, 4B, C). For example, during HCMV infection, Golgi MCS proteins are elevated throughout infection while the overall Golgi proteome decreased in abundance. As an example of distinct temporality, peroxisomal MCS proteins increased in abundance 72 hours prior to the observed global increase in the peroxisomal proteome (Fig. 3C). In contrast, select MCS proteins (e.g., the ER-resident VAP proteins during HCMV infection) appear to change in sync with proteins of related localization, with abundance trends being damped down after comparison to organelle marker proteins (Supplementary Fig. 5A). For the functional comparisons during HCMV infection, nearly all MCS proteins were enriched above proteins of similar functional

pathways. More subtle changes, specific for certain proteins or timepoints, were observed for the other infections examined (Supplementary Figs. 4C, 5B).

Our data point to MCSs that may underlie HCMV-induced organelle remodeling events with mechanisms yet undefined. For example, ER-peroxisome MCS proteins (ACBD5, ACBD4, VAP-A/B) were altered in a virus infection-specific manner, being increased during HCMV infection and decreased during Infl. A infection (Fig. 1D). Peroxisome functions were recently recognized to encompass both anti-viral immune signaling and pro-viral lipid metabolism, and viruses like HCMV and Infl. A manipulate peroxisome dynamics to control the balance of these functions[57]. As the ACBD5-VAPs complex regulates both peroxisome membrane growth and lipid metabolism[58,59], it is possible that ER-peroxisome contact contributes to the distinct peroxisome remodeling events observed during different infections. Another example of a virus-specific trend includes proteins localized to ER-endosome contacts. During HCMV infection, we find PTPN1, which deactivates EGFR at ER-endosome MCSs[60], increasing in sync with EGFR downregulation (Fig. 1C). Given that HCMV inhibits cell-cell signaling by internalizing and degrading EGFR[61,62], HCMV may engage ER-endosome contacts as an additional negative regulator of EGFR function. Additionally, the EGFR pathway is involved in numerous downstream signaling outputs important for HCMV infections, including activation of cell-intrinsic responses[63,64], entry into latency[62,65,66], and trafficking of virus capsids[67]. Although the contribution of ER-endosome contacts to EGFR function during infection remains unexplored, our MCS-PRM data points to the relationship between PTPN1 and EGFR, which requires proximal interactions at ER-endosome interfaces, as a promising avenue for further elucidating the complex EGFR regulation during HCMV infectious cycles. HCMV also increased the levels of ER-endosome proteins that control endosomal cholesterol content and vesicle trafficking (e.g., STARD3/3NL, VAP-A/B) during AC formation and virus assembly (48-120 hpi) (Fig. 1C). In uninfected cells, increasing these proteins results in cholesterol-rich endosomes accumulating near the nucleus, while decreases lead to cholesterol-rich plasma membrane and enhanced retrograde trafficking[68-70]. These changes to secretory organelles are known to occur during HCMV infection[71], yet the mechanisms underlying these endosomal rearrangements remain largely uncharacterized. As HCMV has cholesterol-rich envelopes[72,73], the differential regulation of ER-endosome MCSs may contribute to viral assembly.

Mitochondrial MCS proteins were the most broadly regulated during HCMV infection (up to 5-fold increase, including after normalization to both functional and localization markers) (Fig. 1D, Supplementary Figs. 3-5). In contrast to their increase during HCMV infection, these proteins decreased by up to 2-fold, 3-fold, and 2-fold (after normalization) for HSV-1, Infl. A, and HCoV-OC43 infections, respectively. For HCMV, mitochondrial MCSs begin increasing early (8 hpi) and continue throughout infection, especially for those involved in ER-tethering (e.g., VAP-A/B, MFN1/2), fission (e.g., DNM1L/2, MFF, TBC1D15), mitochondrial membrane integrity (e.g., SAMM50, MTX1/2, MIC60), and ER-mediated calcium transfer (e.g., PTPIP51, VDACs, HSPA9, ITPR3) (Fig. 1C). As HCMV upregulates ER-mitochondria calcium flux, drives mitochondrial fragmentation, and restructures cristae to altogether enhance bioenergetics for virus production[16,74], our findings point to increased ER-mitochondria contact as a possible driver of these remodeling events.

We next tested if viral gene production is necessary for the regulation of MCS proteins during HCMV infection (Supplementary Fig. 6). Specifically, we used both UV-irradiation of viral particles prior to infection—preventing nearly all viral transcription by compromising viral genome integrity—and treatment with a clinical inhibitor of the viral DNA polymerase, phosponoformic acid (PFA, also known as Foscarnet)—preventing expression of late viral gene classes[75]. After confirming the expected inhibition of viral gene production

(Supplementary Fig. 6B), we applied MCS-PRM and found that UV-irradiation reversed the majority of MCS abundance changes characterized for wildtype HCMV infections, with many proteins decreasing in abundance by 24 hpi (Supplementary Fig. 6C, D). This demonstrates that viral gene products are necessary to induce the widespread upregulation of MCS proteins during HCMV infection. Alternatively, inhibition of late viral genes altered MCS changes in a protein-specific manner. For example, PFA treatment prevented the increase in MCS proteins at ER contacts with Golgi, endosome, and PM (e.g., VAP-A/B, ORP5/8, CERT, ESYT1/2, ANXA1/6), while the majority of other trends remained the same as in untreated cells. These MCS proteins control lipid and calcium transfer functions involved in AC formation, which occurs late (>48 hpi) during infection and is known to rely on expression of late viral genes[11]. Our finding that late viral genes are also required for upregulation of these ER-endocytic MCS proteins points to the role of these proteins in generation of the viral AC. Given that UV-irradiation blocked the increase of the majority of MCS proteins during HCMV infection, altogether our findings suggest that immediate early or early viral gene products, such as transcriptional or translational activators, are responsible for the nearly global upregulation of MCS proteins during infection.

**Asymmetric mitochondria-ER encapsulations support infection**

Our MCS-PRM findings enabled us to formulate hypotheses of how MCSs control the progression through different stages of the virus replication cycle. Given the prominent regulation of ER-mitochondria MCS proteins discussed above, we first focused on our observation that HCMV drove the greatest changes in ER-mitochondria MCS protein levels, with average increases of up to 5-fold (Supplementary Fig. 1C, D). This suggested that ER-mitochondria interactions increase during HCMV infection. To test this, we turned to fluorescence confocal microscopy to characterize ER-mitochondria phenotypes as HCMV infection progresses. This required optimization of transient transfections in fibroblasts, which needed to persistently express fluorescent organelle labels for the duration of 120-hour HCMV infections, and sample preparation and fixation amenable to visualizing spatiotemporal organelle interactions in infected cells (see Methods). Confirming adequate progression through the virus replication cycle in fixed cells, we observed hallmark organelle remodeling, including increased ER membrane density (beginning at 24 hpi), kidney-like nuclear envelope reshaping (beginning at 72 hpi and prominent by 96 hpi), and mitochondrial fragmentation (by 48 hpi) (Supplementary Fig. 7A, B).

In uninfected cells through 24 hpi, ER-mitochondria interactions were characterized by ER tubules crossing filamentous mitochondria, often at sites of mitochondrial membrane constrictions (Supplementary Fig. 2A, B). This is well-described for baseline ER-mitochondria interactions in mammalian cells[76–78]. As infection progressed, we identified unique ER-mitochondria contact structures, with mitochondria encapsulated in pockets of ER membranes in two- and three-dimensional space (Fig. 2B, C, Supplementary Fig. 7C, D, Supplementary Movie 1). This type of MCS structure, which we term mitochondria-ER encapsulation (MENC), has not been previously reported in uninfected or HCMV-infected cells. When we quantified ER tubule crossings versus MENCs at each timepoint, we found that MENCs occurred infrequently from 0-24 hpi, while nearly all mitochondria exhibited this phenotype post-72 hpi (Fig. 2D). To verify that these static structures represent membrane contacts, live-cell imaging was used to test whether MENCs are maintained across time and space. In uninfected cells, dynamic ER tubules weave across mitochondria, often changing in location and number of crossings (Fig. 2E, Supplementary Movie 2). We observed similar ER-mitochondria dynamics at 24 hpi, including ER-marked mitochondrial fission events. Beginning at 48 hpi, nearly all mitochondria are encapsulated in ER, maintaining stable contact throughout the movie duration (Fig. 2F). MENCs

exhibited asymmetric spatiotemporal behavior, with ER membranes remaining anchored along the length of an individual mitochondria even during organelle trafficking (Supplementary Movie 2). Using live super-resolution microscopy to further characterize this phenotype, we found that the outer mitochondrial membrane (OMM) increases in proximity to ER membranes as infection progresses (Fig. 2G, Supplementary Movie 2). By 120 hpi, ER-OMM junctions are overlaid along one side of the mitochondrial length, maintaining asymmetric contact across time and space. This MENC phenotype is distinct from previously described ER-mitochondria interactions (tubule crossings at constrictions[76,78], dynamic tubule tracking with mitochondrial ends[77]), representing an as-of-yet unrecognized ER-mitochondria MCS structure induced by virus infection.

We next asked which MCS proteins localize to mitochondria-ER encapsulations. Given the high OMM-ER association, we predicted that MCS protein partners are recruited to these respective locations to facilitate the infection-induced functions of stabilized contact. Among the mitochondrial MCS proteins in our MCS-PRM dataset, VAP-B and PTPIP51 form the most upregulated ER-mitochondria MCS complex during HCMV infection, increasing in tandem by >2 fold (after normalization) (Fig. 3A, Supplementary Fig. 8A). PTPIP51 is a cytosolic protein recruited to ER-mitochondria MCSs by VAP-B, and their tethering interactions are required for ER-mitochondria calcium transfer and mitochondrial integrity[38,79], processes known to be modulated by HCMV[19,74]. Using immunofluorescence (IF), we observed that endogenous PTPIP51 localized diffusely along mitochondria in uninfected cells, concentrating at ER tubule crossings (Fig. 3B, Supplementary Fig. 8B). As infection progressed, PTPIP51 asymmetrically accumulated along the mitochondrial periphery, reflecting OMM-ER association by co-localizing with ER membranes encapsulating mitochondria. VAP-B also re-localized from ER tubule crossings to MENCs, exhibiting asymmetric co-localization with PTPIP51 by late infection (Fig. 3C, Supplementary Fig. 8C-D). VAP-B localization to MENCs was evident by 48 hpi, followed by PTPIP51 at 72 hpi, when asymmetric localization became the primary phenotype for both proteins (Fig. 3D-F). To further interrogate the stability of VAP-B localization, we used live-cell imaging. eGFP-VAP-B became anchored to MENCs, forming fiber-like structures that bridged ER membranes to mitochondria (Fig. 3G). To test whether MENC accumulation involves increased PTPIP51-VAP-B tethering, we used proximity ligation assays (PLA), which measure endogenous protein interactions with spatial sensitivity (≤40 nm), being suited for MCS studies[80]. PTPIP51-VAP-B interactions increased at 72 hpi (Fig. 3H, Supplementary Fig. 9A), in sync with re-localization.

As MENC formation (48 hpi, Fig. 2D) and VAP-B accumulation (48 hpi, Fig. 3F) precede PTPIP51 re-localization, we asked whether PTPIP51 is recruited to ER-mitochondria encapsulations after their formation. We performed siRNA-mediated knockdowns (KDs) of PTPIP51 during HCMV infection, validating KDs and observing little to no off-target or compensatory effects on other MCS proteins (Supplementary Fig. 9B, C). Our results showed that MENCs still form without PTPIP51 (Supplementary Fig. 8E). However, PTPIP51 KD reduced HCMV titers by over 100-fold (Fig. 3I), a defect maintained across the measurement of viral growth curves (72-168 hpi) (Supplementary Fig. 9D), indicating a critical role in virus production. As HCMV assembly begins at 72 hpi, when PTPIP51-VAP-B interactions increase, and we observe PLA signals clustered near the AC (Fig. 3H), our findings suggest that PTPIP51 is recruited to MENCs to engage with VAP-B for functioning in virus assembly. To test this, we quantified viral proteins from each temporal expression class (immediate early, delayed early, and late). PTPIP51 KD had little effect on the levels of immediate early viral proteins, while reducing delayed early and late protein abundances (Supplementary Fig. 9E). Additionally, we observe that mitochondrial fragmentation is not sufficient for driving PTPIP51 re-localization, as drug-induced fragmentation in the absence of infection does not drive similar

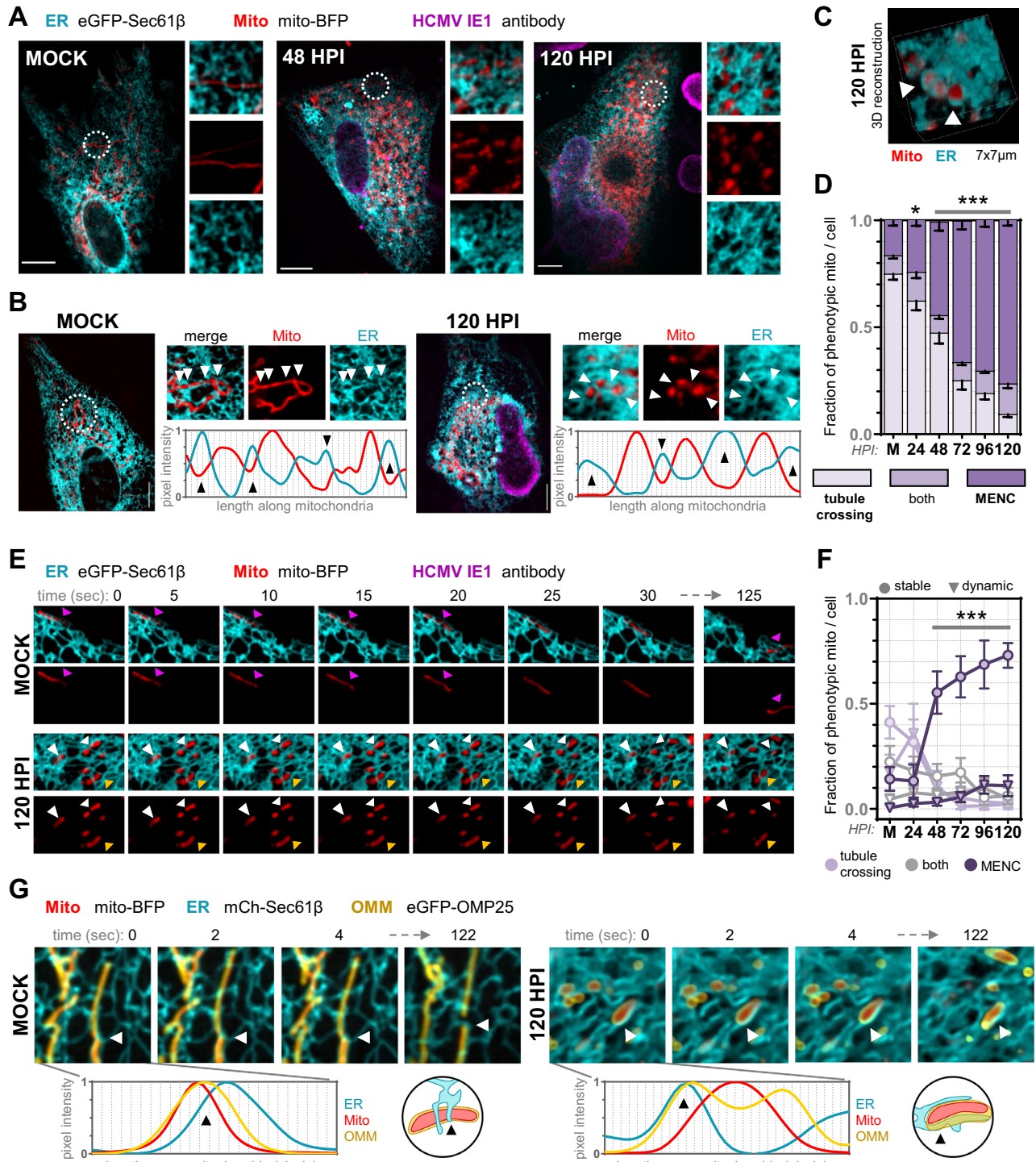

**Fig. 2 | HCMV infection increases and rewires ER-mitochondria contacts into stable encapsulated structures (MENCs). A** Human fibroblasts infected with HCMV and fixed at the indicated timepoints, labeled for: ER (cyan), mitochondria (red), and the viral protein IE1 (magenta) to confirm infection progression. Shown are z-stack max. projections, with a 7×7 μm region (white circles) at right. Scale bars 10 μm. See additional examples in Supplementary Fig. 7. **B** Line-scans of fluorescence intensity along the length of one (Mock, ER tubule crossings) or several (120 hpi, ER encapsulations) mitochondria, with arrows marking ER crossings. **C** 3D reconstruction of a 7×7 μm region from a cell at 120 hpi as in A, with ER membranes encasing mitochondria (MENC). **D** Scoring ER-mitochondria contact phenotypes every 24 hpi from images in A. Key below, error bars are SEM (≥15 mitochondria/cell quantified in ≥33 cells/timepoint, specifically N = 41, 35, 33, 34, 47, and 37 cells for Mock, 24, 48, 72, 96, and 120 hpi, respectively, corresponding to three independent

experiments; *p = 0.0107, ***p ≤ 0.0001 by two-way ANOVA to Mock). **E** Live-cell microscopy (5-sec intervals for 2-min) of ER-mitochondria interactions, shifting from dynamic ER crossings (magenta arrows, *top*) to stable encapsulations (white, *below*) or both (yellow, *below*). **F** Scoring dynamic (triangle) versus stable (circle, maintained for 2 min.) ER contacts on individual mitochondria from movies as in E, categorized as tubule crossings, encapsulations, or both. Key is below, error bars are SEM (≥20 mitochondria/cell for N = 22, 22, 28, 22, 20, 24 cells for Mock, 24, 48, 72 96, and 120 hpi, respectively, corresponding to three independent experiments; ***p ≤ 0.0001 by two-way ANOVA to Mock). **G** Super-resolution movies (2 s. intervals for 2 min., 7×7 μm region) of fibroblasts labeled for: ER (cyan), mitochondria (red), and OMM (yellow). Arrows mark an ER-marked constriction-to-fission event (Mock) or MENC (120 hpi). Line-scans (below) are from the first frame of each movie, across an ER-mitochondria junction.

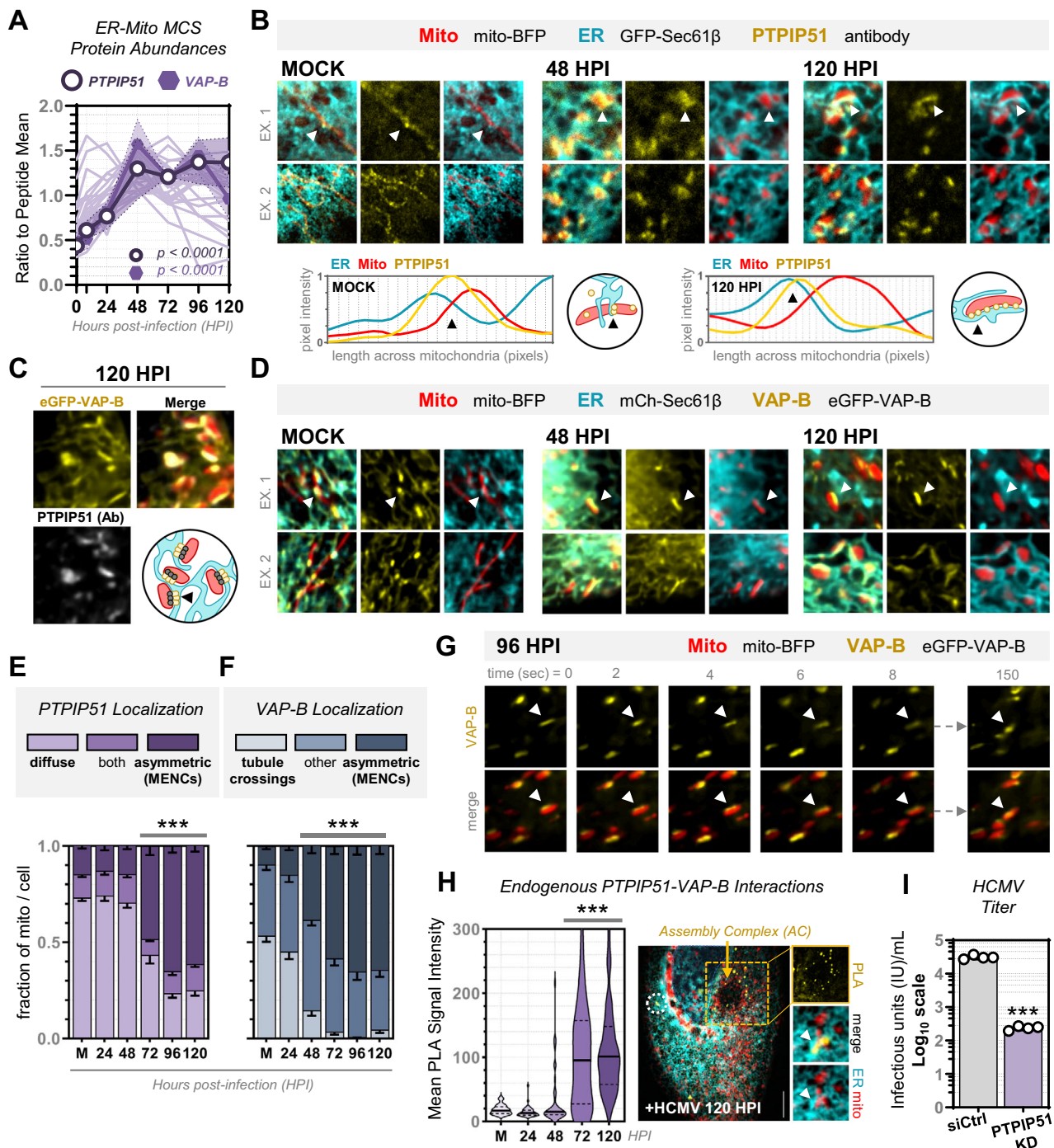

rearrangements (Supplementary Fig. 9F-G). Therefore, we demonstrate that HCMV infection increases and restructures ER-mitochondria interactions into stable encapsulations late in infection, enhancing PTPIP51-VAP-B tethering for the benefit of virus production.

### ER-mitochondria contact enhances STING-mediated signaling

Our results point to a time-sensitive regulation of MCS protein abundances, organelle contacts, and function during HCMV infection. Late in infection, the integration of our MCS-PRM and microscopy analyses led to the discovery that ER-mitochondria contacts increase and are restructured to facilitate virus production. However, early in infection, ER-mitochondria interactions and PTPIP51-VAP-B tethering are not increased (evident in live and fixed-cell analyses up to 24 or 48 hpi, respectively, Figs. 2F, 3G, 4A) despite elevated MCS protein levels.

Additionally, the three other viruses included in our MCS-PRM study decreased ER-mitochondria MCS proteins as early as 2 hpi (Fig. 1D). We sought to understand this perceived conundrum by investigating how time-sensitive regulation of ER-mitochondria contact facilitates HCMV infection. We terminally tethered ER and mitochondria membranes by expressing a synthetic linker (mito-RFP-ER). Mito-RFP-ER reduces MCS distance to ~5 nm and has been used to assess ER-mitochondria contact functions[81,82]. In uninfected cells, tether expression did not perturb mitochondrial structure, cellular viability, or levels of other MCS proteins (Supplementary Fig. 10A-D). Upon HCMV infection, most mito-RFP-ER cells displayed a range of defects in HCMV-induced mitochondrial remodeling, including mistimed fragmentation, aberrant morphology, and juxtanuclear mitochondrial clustering, especially >72 hpi (Fig. 4B, Supplementary Fig. 10E). These phenotypes suggested

**Fig. 3 | The tethering partners VAP-B and PTPIP51 become enriched at MENCs to facilitate HCMV production. A** Average peptide abundances (scaled to the mean) of PTPIP51 (circle) and its ER tether VAP-B (hexagon) during HCMV infection, compared to all other ER-mitochondria MCS proteins (light purple). Shaded regions are standard error of the mean, and p-values are by one-way ANOVA to Mock (data is from MCS-PRM quantification as in Fig. 1, *N* = 6 biological replicates with ≥3 peptides/protein monitored in each replicate, see Supplementary Data 1 for complete list of peptides). **B** IF of endogenous PTPIP51 (yellow) during HCMV infection, showing regions from two different cells for each timepoint. Line-scans (below) are across ER (cyan) and mitochondria (red) junctions (arrows). Shown are 7×7 µm regions. **C** Fixed images (7×7 µm) from cells labeled for mitochondria (red), endogenous PTPIP51 (grey), and VAP-B (yellow) late in HCMV infection. Cartoon represents the observed co-localization of PTPIP51 and VAP-B at MENCs. **D** Images from live cells labeled for VAP-B (yellow), ER (cyan), and mitochondria (red). *Top*, 7×7 µm stills from two different cells for each timepoint; *Lower*, 2-second intervals from the region in C (2.5-minute movie, timepoints *above*). Arrows indicate points of VAP-B accumulation at ER-mitochondria contacts. **E, F** Scoring PTPIP51 and VAP-B localization from images in B and D (≥20 mitochondria/cell for PTPIP51 *N* = 59, 35, 43, 35, 21, 43 cells corresponding to three independently collected experiments and VAP-B *N* = 35, 18, 35, 33, 21, 31 cells corresponding to two independently collected experiments in Mock, 24, 48, 72, 96, and 120 hpi, respectively; error bars are SEM; ***$p \leq 0.0001$ by two-way ANOVA to Mock). **G** Stills (7×7 µm) from live-cell images of cells labeled for mitochondria (*red*, mito-BFP) and VAP-B (*yellow*, eGFP-VAP-B) at 96 hours post-HCMV infection. Arrows indicate stable VAP-B fibers along the mitochondrial length. Timepoints are indicated *above*. **H** Proximity ligation assay (PLA) of endogenous PTPIP51-VAP-B interactions during HCMV infection. *Left*, Violin plots of PLA intensity (solid line is mean, dotted lines are quartiles, *N* ≥ 38 cells/timepoint, ***$p \leq 0.0001$ by two-tailed student's t-test to Mock). *Right*, Image of the PLA (yellow) analysis at 120 hpi, with regions near the viral AC (*top*) and ER-mitochondria co-localization (*lower*). More examples shown in Supplementary Fig. 9. **I** Virus titers of PTPIP51 KD versus siRNA control, collected at 120 hpi (Log$_{10}$ scale, *N* = 4, ***$p \leq 0.0001$ by two-tailed student's t-test to siCtrl).

that virus-driven mitochondrial dynamics are broadly disrupted by prematurely increased ER contact. Indeed, virus titers exhibited a tether concentration-dependent decrease of up to 10-fold (Fig. 4C), demonstrating that ER-mitochondria contact is disrupted early and increased late during infection for the benefit of the virus.

We next investigated why ER-mitochondria tethering is inhibited by HCMV early in infection. As mito-RFP-ER can activate apoptosis under stress[81], we first examined whether ER-mitochondria tethering would overcome viral suppression of apoptosis. Measurement of apoptotic cell populations (TUNEL assays) and PARP cleavage (apoptosis marker) did not correlate with tether expression before or during HCMV infection (Supplementary Fig. 11A-B). We next considered that both organelles coordinate immune signaling, with the immune factors MAVS and STING primarily localized to mitochondria and ER, respectively[83]. Several HCMV proteins are known to target and inhibit STING[6,84–88]. Our microscopy data showed that ER-mitochondria tethering reduced expression of the immediate early viral protein 1 (IE1), a read-out of active HCMV infection[89], both in total IE1-producing cells and IE1 levels (Fig. 4D-E). Further MS investigation revealed that the levels of viral proteins from each temporal expression class were reduced through 48 hpi (Fig. 4F), indicating delayed infection onset.

To examine if anti-viral signaling underlies these observations, we used IF to monitor STING activation early during HCMV infection, quantifying endogenous STING re-localization to cytosolic signaling aggregates[90,91]. ER-mitochondria tethering enhanced STING re-localization at 6 hpi, 24 hpi, and upon transfection with vaccinia virus DNA (VACV 70mer), a known STING stimulator (Fig. 4G-H). STING aggregates were observed in both tether-expressing and neighboring cells, indicating cytokine secretion. To test this, we first monitored activation of the downstream signaling partners TBK1 and IRF3, which are phosphorylated by STING after re-localization[91,92]. Western blot analysis showed increased TBK1 and IRF3 phosphorylation through 48 hpi (Supplementary Fig. 11C-F). We then measured interferon secretion via fluorescent bead assay, finding that ER-mitochondria tethering increases the concentration of secreted interferons from α, β, and γ classes, both prior to and during virus infection (Fig. 4I). Finally, we measured virus production in tether-expressing cells treated with known STING and TBK1 inhibitors, H-151[93] and GSK8612[94]. STING and TBK1 loss-of-function resulted in a partial rescue of virus titers (Fig. 4J, Supplementary Fig.11G-H). The remaining defects in virus production are indicative of the above mentioned complexity of mitochondrial alterations caused by mito-RFP-ER upon infection (Supplementary Fig. 10E), and the possibility of STING performing both pro- and anti-viral roles during HCMV infections[6,85,95]. Together with our other findings that mito-ER tethering causes STING relocalization, TBK1 and IRF3 phosphorylation, cytokine secretion, reduced virus gene expression, and delayed infection onset, our

results demonstrate that the STING immune pathway is a contributor to the anti-viral effect of premature ER-mitochondria tethering.

Given the broad importance of the STING pathway, we next tested the anti-viral capacity of increased ER-mitochondria tethering in HSV-1 and Infl. A infections. In both cases, mito-RFP-ER expression decreased virus production (up to 8- and 12-fold, respectively) (Fig. 4K-L). STING re-localization also increased by 4 hours post-HSV-1 infection in tether-expressing cells (Fig. 4M). Together with our findings that both HSV-1 and Infl. A reduce ER-mitochondria MCS protein abundances (by up to 2-fold and 3-fold after normalization, respectively) (Fig. 1D) and that HCMV decreases ER-mitochondria interactions early in infection (Fig. 4A), our results suggest that the time-sensitive regulation of ER-mitochondria contacts may modulate STING signaling during infection.

## ACBD5-mediated ER contacts increase to remodel peroxisomes

Our discovery that the structure, function, and extent of ER-mitochondria contact contributes to pro-viral organelle remodeling prompted us to further explore our MCS-PRM dataset. Peroxisome-ER contacts were prominently regulated by HCMV infection. All four proteins at the ER-peroxisome MCS (ACBD4, ACBD5, VAP-A, VAP-B) increased in abundance throughout HCMV infection (up to 3-fold after normalization), in contrast to their slight decreases during the other infections examined (e.g., 1.5-fold, 2-fold, and 1.5-fold decreases after normalization in HSV-1, Infl. A, and HCoV-OC43 infections, respectively) (Fig. 1D). Peroxisomes have recently gained attention as organelles that pivot between anti- and pro-viral roles during virus infections, with functional alterations often involving changes to peroxisome shape, composition, and numbers[57]. Although the ER is central to the regulation of peroxisomes, ER-peroxisome contacts have not been previously examined during virus infections. Our MCS-PRM showed that, by 48 h post-HCMV infection, the primary tethering partners ACBD5 and VAP-B are increased >threefold (Fig. 5A). This MCS partnership controls peroxisome membrane expansion and the peroxisome-to-ER transfer of plasmalogens[58,59]. Given that HCMV induces peroxisome enlargement and plasmalogen synthesis for virion assembly[17], we hypothesized that ACBD5-mediated ER-peroxisome contact increases during infection and represents a mechanistic basis for pro-viral peroxisome remodeling.

Using PLA, we tested whether increased ACBD5 and VAP-B abundances reflect enhanced organelle tethering. PLA puncta quantification across the HCMV replication cycle revealed increased ACBD5-VAP-B interactions, especially after 48 hpi, indicating amplification of ER-peroxisome tethering (Fig. 5B, Supplementary Fig. 12A). To further characterize ER-peroxisome interactions, we turned to live-cell imaging. In uninfected cells, as expected[58,59], spherical peroxisomes are associated with ER membranes, often

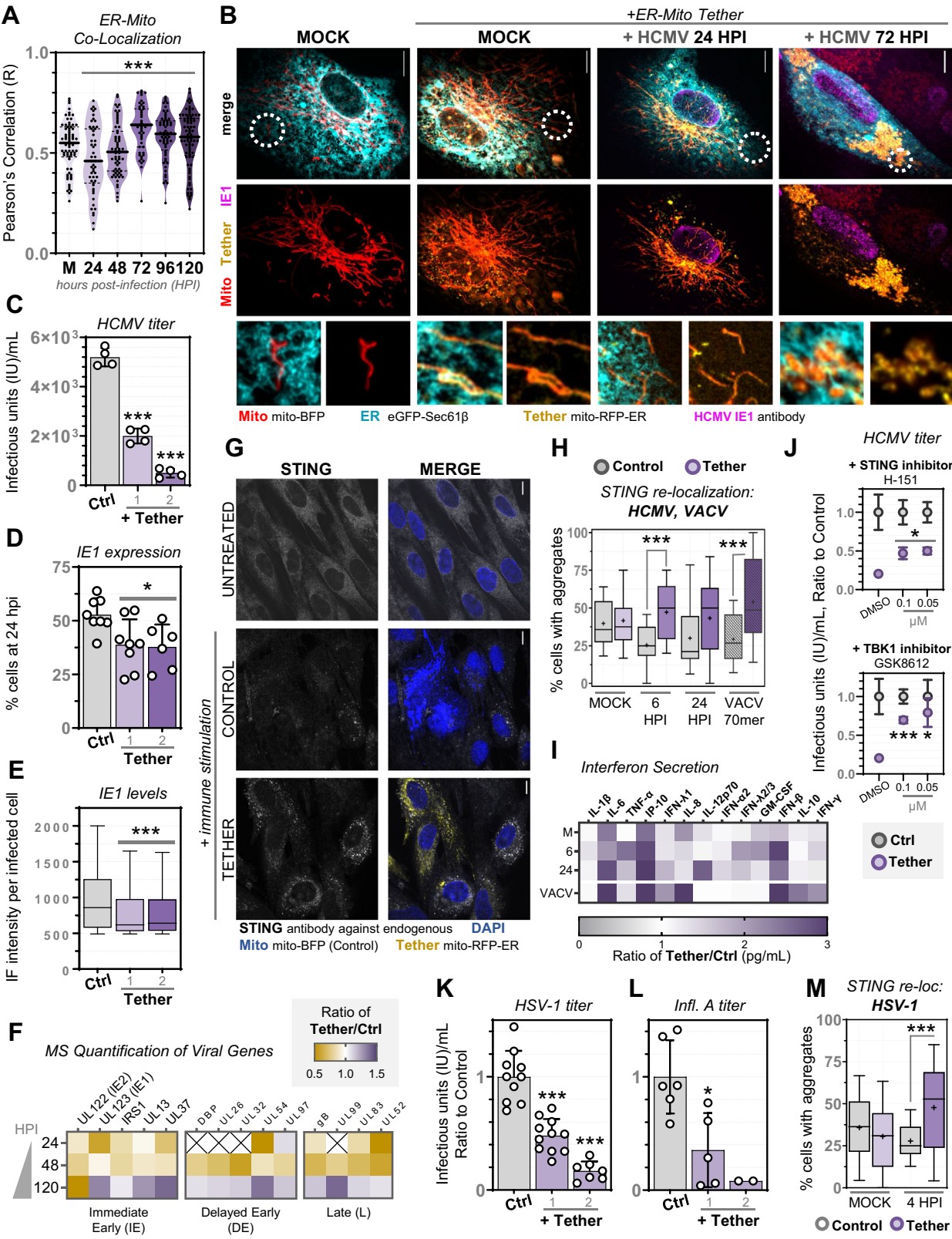

exhibiting dynamic behavior by trafficking along ER tubules (Fig. 5C). As the infection progressed, peroxisomes remained anchored to the ER, moving together in time and space, and became increasingly static in movement (Supplementary Movie 3, Supplementary Fig. 12B-C). As restricted peroxisome motility represents increased ER tethering[59], we used particle tracking to quantify this phenotype. Peroxisomes decreased in movement from 48-120 hpi.

(Supplementary Fig. 12D), coinciding with elevated ACBD5-VAP-B abundances and tethering interactions.

Infection-induced enlarged peroxisomes, present by 72 hpi, exhibited particularly extensive plastering on ER. In every live-cell movie we captured, nearly all enlarged peroxisomes maintained static contact with expanded ER membranes (Fig. 5C, Supplementary Fig. 13A). Further IF analysis of fixed cells, labelled for ER and

**Fig. 4 | ER-mitochondria tethering enhances STING-TBK1-IRF3 immune activation upon HCMV infection. A** ER-mitochondria co-localization from images in Fig. 2A (line at median, $N = 71, 51, 64, 49, 60$, and 86 cells/timepoint in Mock, 24, 48, 72, 96, 120 hpi, respectively, corresponding to 3 independent experiments; ***$p \leq 0.0001$ by one-way ANOVA to Mock). **B** Images of fibroblasts expressing mito-RFP-ER (Tether, yellow) and labeled for ER (cyan), mitochondria (red), and IE1 (magenta). White circles indicate zoomed regions *below*. Scale bars 10 μm. **C** HCMV titers from control versus Tether cells (1: 500 ng, 2: 750 ng) ($N = 4$, ***$p \leq 0.0001$ by two-tailed student's t-test to control). **D** IE1 expressing cells (immunofluorescent focus forming assay) at 24 hpi, comparing Tether (1: 500 ng, 2: 750 ng) to control (Ctrl, mito-BFP) ($N = 8$ biological replicates for Mock and 500 ng, $N = 6$ for 750 ng; *$p = 0.014$ by two-tailed student's t-test to Mock). **E** IE1 levels (immunofluorescent intensity) per nuclei, comparing cell populations as in **D** (Tukey box-and-whisker plot with lines at median; $N = 1342, 1628, 1247$ cells in Ctrl, 500 ng, 750 ng, respectively; ***$p \leq 0.0001$ by one-way ANOVA with Dunnett's multiple comparisons test to Ctrl). **F** Targeted MS quantification of viral protein abundances from different temporal expression classes (IE, DE, L). Heatmap key at *top*, timepoints *left*. An 'X' indicates proteins not detected in either condition.

**G** Immunofluorescence of endogenous STING localization, comparing non-transfected, control (mito-BFP), and mito-RFP-ER (tether) cells. DAPI and mito-BFP excite at 405 nm and are both shown in blue. Scale bars 10 μm. **H** Scoring STING aggregation during HCMV infection (Mock, 6, 24 hpi), and upon VACV 70mer transfection. Plotted are percent aggregate-positive cells per field of view (for Mock, 6 hpi, 24, hpi, VACV conditions, respectively: $N = 167, 118, 100, 103$ cells/condition for Ctrl, $N = 160, 188, 80, 73$ cells/condition for Tether; line at median, + at mean, whiskers min-max, ***$p \leq 0.001$ by two-tailed student's t-test to Ctrl). **I** Quantification of secreted interferon abundance comparing conditions as in **H** by fluorescent bead assay. Shown as a ratio of tether to control cells, key is *below* ($N = 2$). **J** HCMV titers from control (*grey*) and tether (*purple*) cells treated with STING inhibitor H-151 (*top*) and TBK1 inhibitor GSK8612 (*lower*). Shown as the average ratio to control for each condition ($N = 4$, error bars are standard deviation; *$p \leq 0.05$, ***$p \leq 0.001$ by two-tailed student's t-test to DMSO). **K, L** HSV-1 ($N \geq 7$) and Infl. A ($N \geq 2$) titers from control versus tether cells (1: 500 ng, 2: 750 ng) (*$p \leq 0.05$, ***$p \leq 0.001$ by two-tailed student's t-test to control). **M** Scoring STING aggregates during HSV-1 infection ($N \geq 252$ cells/timepoint), box-and-whisker plot as in **H** (***$p \leq 0.001$ by two-tailed student's t-test to Ctrl).

peroxisome membranes (PEX14 antibody), revealed that enlarged peroxisomes preferentially localized to the densest areas of ER at every timepoint (Fig. 5D). Peroxisome membranes spread along ER in irregular structures in three-dimensional space (Supplementary Movie 4). This was not observed for spherical peroxisomes in uninfected cells, or for fragmented peroxisomes at 72–120 hpi, which remained associated with discrete ER tubules and had less ACBD5 co-localization than their enlarged counterparts (Supplementary Fig. 13B-C). Therefore, we confirm that the changes in protein abundance uncovered by MCS-PRM represent increased ER-peroxisome contact during HCMV infection, and further demonstrate the enrichment of these contacts at infection-derived enlarged peroxisomes.

We next asked whether increased ER-peroxisome contacts function for the benefit of virus or host. We optimized ACBD5 siRNA-mediated KDs and plasmid-based overexpressions (OEs) in fibroblasts, confirming expression throughout infection and low cytotoxicity (Supplementary Fig. 13D-F). We expected that ACBD5 genetic manipulations would prevent (KDs) or enhance (OEs) pro-viral alterations to peroxisome structure. Using IF, we observed peroxisomes in uninfected ACBD5 KD cells to have reduced surface area and volume compared to control (Fig. 5E-F, Supplementary Fig. 13G-H). By 120 hpi, ACBD5 KD prevented the HCMV-induced remodeling of peroxisome morphology, such as the formation of enlarged peroxisomes observed in control cells (Supplementary Movie 5). Specifically, peroxisomes in ACBD5 KDs remain the same size and spherical shape throughout infection. In contrast, ACBD5 OE increased peroxisome size even in uninfected cells in a concentration-dependent manner, and at 120 hpi nearly all peroxisomes exhibited enlarged and deformed membranes (Fig. 5F, Supplementary Movie 5). These results demonstrate that ACBD5-mediated ER contacts, especially those formed by enlarged peroxisomes along ER membranes (72-120 hpi), are both sufficient and required for virus-directed peroxisome membrane expansion.

Given this role of ACBD5 in peroxisome restructuring during infection, we next tested its impact on virus production. ACBD5 KD decreased HCMV titers by nearly 14-fold (Fig. 5G). This impact was similar to KD of GNPAT (Supplementary Fig. 13I), the peroxisomal initiator of plasmalogen synthesis previously shown to be essential for HCMV replication[17]. However, we also observed that ACBD5 OE reduces HCMV production by half (Fig. 5G). This pointed to an anti-viral function for ACBD5 OE, a result we initially found puzzling as ACBD5 increases in abundance and tethering function during infection. Reconsidering our microscopy data, we noticed that ACBD5 OE prevented infection-induced peroxisome biogenesis, as cellular peroxisome numbers remained steady through 120 hpi in comparison to a five-fold increase in control cells (Fig. 5H). In contrast, ACBD5 KD cells exhibited greater peroxisome numbers in both uninfected and infected conditions, suggesting a previously unrecognized role for ACBD5

as a negative regulator of peroxisome proliferation. As we observed ER-peroxisome tethering interactions to increase after 24 hpi (Fig. 5B), while peroxisome numbers increase by 8 hpi[17,96], our results indicate that ER contact with peroxisomes is suppressed early and increased later in HCMV infection to control the balance between pro-viral peroxisome biogenesis and membrane expansion.

We next assessed the impact of ACBD5 on the other human viruses included in our MCS-PRM study. ACBD5 OE reduced the titers of HSV-1, Infl. A, and HCoV-OC43, with the most pronounced effect on coronavirus production (nearly 1000-fold) (Fig. 5I-K). Peroxisome structure-function relationships have not been examined in influenza or HCoV-OC43 infections. For HSV-1, we previously showed that peroxisome numbers increase during infection[17]. Our findings that ACBD5 exerts an anti-viral effect on these tested infections raises the possibility of a role for ER-peroxisome MCSs in supporting virus production, which remains to be examined.

## Discussion

The structure-function relationships of organelles across sub-cellular space and biological time can shift the balance between cellular health and disease. By linking organelles in dynamic intra-cellular networks, membrane contact sites (MCSs) are positioned to finely tune organelle localization, morphology, molecular composition, and functional capacity[31]. As disruption of these features underlies cellular pathologies ranging from neurodegenerative to metabolic disorders, viewing organelle remodeling through the lens of MCSs is a powerful strategy for elucidating the global fingerprints of diseases. Given the remarkable rate of discovery for MCS proteins and functions, tools that can simultaneously define the spatial, temporal, and functional components of MCSs in complex biological systems are needed.

Here, we report the design, application, and validation of an MCS-PRM experimental platform for the targeted yet high-throughput profiling of global MCS regulation during dynamic biological processes. This assay monitors the abundances of functionally-defined MCS proteins across all major organelles as a read-out for changes in inter-organelle contacts and functions. By performing precise and sensitive quantifications of signature peptide markers, this assay is suitable for investigating complex systems—such as clinical samples or time-sensitive treatments—as it requires little starting material and is adaptable to other cell types, tissues, and biological conditions. MCS proteins can be multi-localized, multi-functional, and contribute to multiple organelle interfaces (e.g., the master ER tethers VAP-A/B). Therefore, this assay can uncover the regulation of MCS proteins during a given biological process, helping to formulate functional hypotheses of multi-faceted organelle regulation. Additional validation using microscopy or proximity ligation assays, as shown in this study, may be

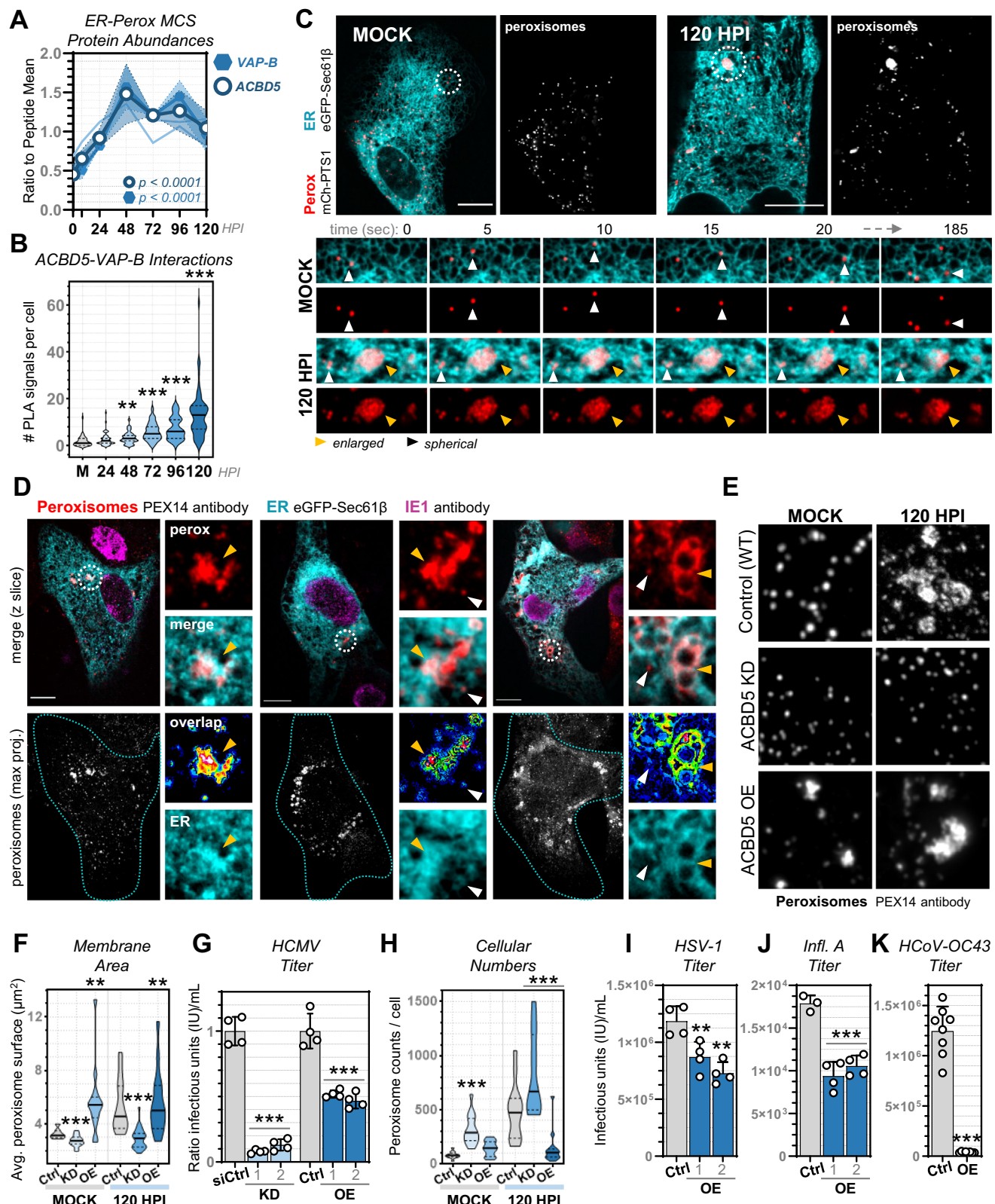

needed to support the PRM data and gain specific insight into the localization of the tethering event. Indeed, we leverage MCS-PRM to define the temporal rewiring of organelle contacts by HCMV, and compare these trends to infections with other human viruses (HSV-1, Infl. A, and HCoV-OC43). We go on to discover a previously unreported contact structure formed by infection (MENCs), identify a role for ER-mitochondria tethering in activating STING immunity, and demonstrate

that ER contacts with peroxisomes provide the mechanistic underpinnings for pro-viral peroxisomal remodeling.

We discover virus-specific alterations to MCS protein abundances (Fig. 1). HCMV drives a global upregulation of MCS protein levels, reflecting the ability of this virus to remodel every major organelle during its infectious cycle. Our findings during HCMV infection lead us to propose a model whereby temporal control of MCS protein

**Fig. 5 | ACBD5-mediated ER-peroxisome contacts underlie HCMV-driven changes to peroxisome size and numbers. A** Average peptide abundances (scaled to mean) of ACBD5 (circle) and its ER tether VAP-B (hexagon) during HCMV infection, compared to ACBD4 and VAP-A (light blue lines). Shaded regions are SEM, and *p* values are by one-way ANOVA to Mock (data is from MCS-PRM quantification as in Fig. 1, *N* = 6 biological replicates with ≥3 peptides/protein monitored in each replicate, see Supplementary Data 1 for complete list of peptides). **B** PLA quantification of endogenous ACBD5-VAP-B interactions. Plotted are PLA signal counts (solid line at mean, dotted lines at quartiles, *N* ≥ 40 cells/timepoint, **$p \leq 0.01$, ***$p \leq 0.001$ by two-tailed student's t-test to Mock). **C** Movies of ER (cyan) and peroxisomes (white/red) before and 120 hpi, showing whole-cell and zoomed stills (white circles, lower). Scale bars 10 μm. **D** Fixed fibroblasts in late (72–120 hpi) stages of HCMV infection, labelled for: ER (cyan), peroxisome membranes (red), and HCMV IE1 (magenta). Each channel from a zoomed region (white circles) is shown at right, including an ER-peroxisome overlap mask heat-colored by increasing overlap. Arrows indicate enlarged (yellow) or small (white) peroxisomes, localizing with expanded or tubular ER, respectively. Scale bars 10 μm. **E** IF analysis of peroxisomes (PEX14 antibody) before and 120 hpi, comparing control to ACBD5

KD and OE (10×10 μm). See Supplementary Fig. 13 for more examples. **F** Quantification of peroxisome surface area in control (Ctrl, *N* = 8973 peroxisomes from 15 cells in Mock, *N* = 11,991 peroxisomes from 15 cells in 120 hpi), ACBD5 KD (*N* = 16,625 peroxisomes from 15 cells in Mock, *N* = 33,392 peroxisomes from 28 cells in 120 hpi), and ACBD5 OE (*N* = 3961 peroxisomes from 15 cells in Mock, *N* = 5174 peroxisomes from 15 cells in 120 hpi) cells before and 120 hpi (solid line at median, dotted lines at quartiles; **$p \leq 0.01$, ***$p \leq 0.001$ by two-tailed student's t-test to Ctrl/timepoint). **G.** HCMV titers from ACBD5 KDs (two siRNAs) and OEs (1: 250 ng, 2: 500 ng) versus either siRNA or plasmid controls (*N* = 4, ***$p \leq 0.001$ by two-tailed student's t-test). **H** Peroxisome counts per cell in control (Ctrl, *N* = 70 cells in Mock, *N* = 20 cells in 120 hpi), ACBD5 KD (*N* = 28 cells in Mock, *N* = 23 cells in 120 hpi), and ACBD5 OE cells (*N* = 42 cells in Mock, *N* = 15 cells in 120 hpi) (solid line at median, dotted lines at quartiles; ***$p \leq 0.0001$ by two-tailed student's t-test to Ctrl/timepoint). **I–K.** HSV-1 (*N* = 4 biological replicates, *$p$ = 0.0151 and **$p$ = 0.0014), Infl. A (*N* ≥ 3, ***$p$ = 0.0005 and ***$p$ = 0.0003), and HCoV-OC43 (*N* = 8, ***$p \leq 0.0001$) titer measurements in control versus ACBD5 OE cells (1: 250 ng, 2: 500 ng, p-values by two-tailed student's t-test to control).

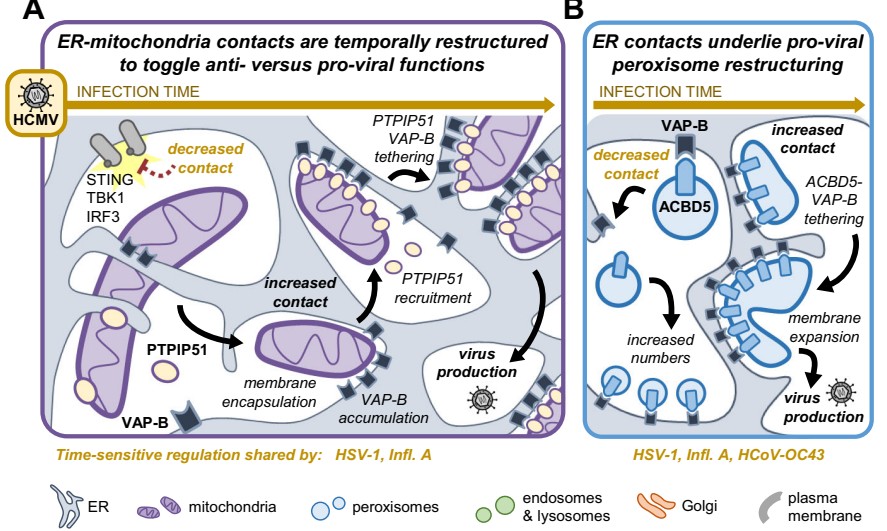

**Fig. 6 | ER-mitochondria and ER-peroxisome contacts are required for HCMV-driven organelle remodeling. A** Model for ER-mitochondria contact remodeling during the HCMV infectious cycle. Contact is decreased early to evade STING-TBK1-IRF3 signaling induced by increased ER-mitochondria tethering, a shared function with HSV-1 and likely Infl. A. Late in infection, contact is increased and restructured into ER-mitochondria encapsulations (MENCs) that recruit VAP-B and then PTPIP51, which increase in tethering interactions, and PTPIP51 is required for HCMV production. **B** Model for ER-peroxisome contact functions during HCMV infection. Early in infection, increased ACBD5-mediated contact prevents virus-induced peroxisome proliferation and restricts HCMV, HSV-1, Infl. A, and HCoV-OC43 production. As infection progresses, ER-peroxisome contact is increased and enriched at the enlarged peroxisomes formed by infection, which require ACBD5-VAP-B tethering to form.

abundances, structures, and tethering interactions provide the means to toggle between anti- and pro-viral organelle functions (Fig. 6). During HCMV infection, ER-mitochondria contact is restructured with temporal sensitivity to facilitate immune evasion and the progression through infection (Fig. 6A). Early in infection, ER-mitochondria interactions are decreased to circumvent STING-TBK1-IRF3 anti-viral signaling.

Late in HCMV infection, increased MCS protein abundances and tethering interactions are linked to the formation of an ER-mitochondria encapsulation structure (MENC). MENCs asymmetrically accumulate VAP-B and PTPIP51, which is required for virus production. At the peroxisome, ER contacts facilitate the HCMV-driven remodeling of peroxisome morphology and numbers (Fig. 6B). During HCMV infection, ACBD5-mediated ER contacts with peroxisomes are suppressed early and increased late in infection to toggle the balance between peroxisome proliferation and membrane enlargement for the viral benefit. ACBD5 abundance also restricted the production of HSV-1, Infl. A, and HCoV-OC43, pointing to a possible broad role for ER-peroxisome contacts in virus replication, which remains to be further explored.

Subcellular control of the metabolism-immune signaling axis has emerged as a determinant of the biology and pathogenesis of infections, with mitochondria and peroxisomes at the center of this axis[57,97]. Both organelles facilitate functions that can favor either the host or virus, being home to anti-viral sensors (e.g., STING at mitochondria and MAVS at both[91]), stress response pathways, and cellular metabolism. Viruses face the challenge of selectively inhibiting or enhancing these functions to benefit their replication cycles. As an added conundrum, several viruses drive mitochondria and peroxisome fragmentation to inhibit immune responses, while still maintaining or elevating their metabolic activity[98]. Our findings here show that HCMV and Infl. A infections drive opposite changes in the levels of MCS proteins key to mitochondrial membrane integrity, signaling capacity, and ER tethering (Fig. 1C). This is reminiscent of the differential regulation of mitochondria during these infections, whereby both viruses employ viral proteins to cause fragmentation and disrupt apoptotic/immune signaling, yet decrease (Infl. A) or enhance (HCMV) overall metabolic output[16,23,43,99,100]. Given that ER contacts coordinate the structure-function relationships of mitochondria, Infl. A may

downregulate MCS proteins to disrupt ER contact and destabilize overall mitochondrial biology. In contrast, our findings indicate that by upregulating many of the same proteins HCMV induces specific MCSs to bypass the functional defects otherwise caused by aberrant fragmentation. We uncover the HCMV infection-specific formation of MENCs, which are enriched in PTPIP51 tethered to VAP-B (Figs. 2–3). The PTPIP51-VAP-B complex inhibits autophagy and controls calcium flux for mitochondrial metabolic function[79,82]. As we find PTPIP51 to be required for virus production, our findings point to restructured ER-mitochondria contacts as a strategy by which HCMV maintains a population of fragmented mitochondria still capable of meeting the metabolic demands of infection. It is tempting to speculate that other biological conditions characterized by altered ER-mitochondria contact may similarly form MENCs, such as those involving changes in ER-mitochondria calcium flux (e.g., neurodegenerative disorders[39,101,102]).

Our investigations also illustrate the importance of time-sensitive control of MCSs during infection. During HCMV infection, we find that both PTPIP51 and ACBD5 increase in abundance, are enriched at MCSs, have elevated ER tethering, and are required for virus production. However, we also determine that premature induction of ER contact with either mitochondria or peroxisomes negatively impacts virus replication. At mitochondria, we show that this anti-viral effect is due to enhanced activation of the STING-TBK1-IRF3 immune axis, preventing the establishment of infection (Fig. 4). At peroxisomes, we find that ACBD5 is positioned at a tipping point between peroxisome numbers and structural alterations, whereby prematurely increased ER-peroxisome contact inhibits pro-viral peroxisome proliferation (Fig. 5). Disrupting the temporal regulation of these ER-mediated contacts reduced the titers of each virus we examined. HCMV increased organelle tethering late in infection and required MCS proteins for virus production. This fits with the upregulated MCS protein abundances defined by MCS-PRM, while also indicating the presence of as-of-yet undiscovered viral mechanism(s) for circumventing increased inter-organelle contact early in infection. Virus-host interactions that change the localization or post-translational modifications of MCS proteins, such as the recently identified phospho-FFAT motif for VAP-mediated tethering[52], are promising possibilities that could be explored by coupling MCS-PRM to protein interaction studies or expanding the assay to include modified peptides, such as by phosphorylation. Given the broad relevance of STING signaling and that numerous viruses modulate peroxisome shapes and numbers (e.g., herpesviruses, coronaviruses, human immunodeficiency virus (HIV)[29,57,103]), our findings suggest that the temporal regulation of MCSs can provide a balancing point between the anti- versus pro-viral functions of organelles.

Viruses possess a remarkable ability to reprogram cells in short timescales and with limited gene products. Our work shows that HCMV infection modulates MCS protein abundances and drives the formation of specialized contact structures, including the temporal upregulation of mitochondria-ER encapsulations (MENCs). Hence, MCSs can serve as platforms by which viruses regulate inter-organelle communication to exert functions that span subcellular space and finely tune these functions across infection time. We expect that continued investigations of the virus-driven alterations in MCSs defined here will spur numerous discoveries of how viruses coordinate both the intra- and inter-cellular signaling events that dictate the progression of infection and its resulting pathologies. We also anticipate that our MCS-PRM assay will be leveraged to define global fingerprints of organelle remodeling fundamental to many biological processes, model systems, and human diseases.

## Methods
### Cell culture, virus strains, and infection protocols
All experiments were performed in MRC5 human fibroblast cells (ATCC, CCL-171), except for experiments specific to HCoV-OC43,

which were performed in RPTE epithelial cells (Lonza, CC-2553) and LLC-MK2 cells (ATTC, CCL-7). MRC5 and LLC-MK2 cells were grown as monolayers in Dulbecco's Modified Eagle Medium (DMEM, high-glucose) supplemented with 10% fetal bovine serum (FBS) and 1% penicillin/streptomycin at 37 °C in a 5% $CO_2$ atmosphere. RPTE cells were grown as a monolayer in Lonza Renal Epithelial Basal Medium (REBM, cc-3191) supplemented with REGM Singlequots aliquots (cc-4127) at 37 °C in a 5% $CO_2$ atmosphere.

Human virus strains used in this study are as follows: HCMV AD169 (gift from Dr. Thomas Shenk, Princeton University), HSV-1 17 + (gift from Dr. Beatte Sodeik, Hannover Medical School), Influenza A H1N1 A/Puerto Rico/8/1934 (gift from Dr. Thomas Shenk, Princeton University), and HCoV-OC43 VR-1558 (gift from Dr. Kevin Harrod, University of Alabama at Birmingham; originally purchased from ATCC). HCMV was propagated in MRC5 cells, HSV-1 in U2OS cells, Infl. A in MDCK cells, and HCoV-OC43 was propagated in LLC-MK2 or RPTE cells. HCMV and HSV-1 were purified and concentrated via ultracentrifugation on a 20% sorbitol or 10% ficoll cushion, respectively. Infl. A was not concentrated. HCoV-OC43 was purified and concentrated by ultracentrifugation. All virus stocks were stored at -80 °C until use.

For all infections in this study, cells were grown to 90-100% confluency prior to infection, and inoculations were performed in half the standard media volume. HCMV was inoculated on cells for 1 hour in 10% FBS + DMEM, and then cells were rinsed in warm phosphate-buffered saline (PBS) and left in 10% FBS + DMEM until sample collection. For HSV-1 infections, cells were infected for 1 hour in 2% FBS + DMEM, rinsed in warm PBS, and returned to 10% FBS + DMEM until sample collection. Infl. A infections were carried out in Opti-MEM (OMEM) supplemented with 0.25 µg/mL of TPCK trypsin (Thermo-Fisher Scientific, #20233), with a 1 hour inoculation, warm PBS rinse, and return to OMEM without (PRM-MS) or with (virus titering) TPCK trypsin until sample collection. HCoV-OC43 infections were performed in basal cell media (REBM or DMEM) without FBS by inoculating virus on cells for 1 hour, washing in PBS after removing inoculant, and replacing cells in either REBM or 10% FBS + DMEM until sample collection for RPTE and LLC-MK2 cells respectively.

The multiplicity of infection (MOI) varied by experiment, as indicated in the text or figures. For MCS-PRM, virus MOIs were: HCMV MOI 3, HSV-1 MOI 10, Infl. A MOI 0.8, HCoV-OC43 MOI 5, MOIs selected based on their use in previous proteomic workflows for these viruses. For fluorescent imaging and virus titering during HCMV infections, cells had been transiently transfected (see Methods below) causing elevated immune signaling, so a slightly higher MOI of 5 was used, determined by the lowest MOI that produced minimum 80% infected cells mid-way through infection in control cells, and were kept standard for all genetic conditions for each experiment.

### Sample preparation for MS analyses
At the indicated timepoints for each infection, cell lysates were collected from 2 mL wells or 6 cm cell culture dishes by rinsing and scraping cells into cold PBS, pelleting at 500x $g$ for 3 minutes in a tabletop centrifuge, and freezing cell pellets at -80 °C until analysis. Immediately preceding analysis, cell pellets were lysed in prewarmed 5% SDS lysis buffer containing 100 mM Tris-HCL (pH 7.4), 0.5 mM EDTA, and 100 mM NaCl. Protein concentration was determined by BCA assay (Pierce), and 30 µg of protein was prepared for MS analysis. Samples were reduced and alkylated (25 mM TCEP and 50 mM chloroacetamide) at 70 °C for 20 min, then acidified to 1.2% phosphoric acid prior to protein extraction. S-Trap Micro Spin Columns (Protifi) were used for protein extraction, trypsin digest (using a 1:25 ratio of trypsin to sample protein for 1 h at 47 °C), and sample desalting according to the manufacturer's protocol, apart from performing 5 total wash steps prior to trypsinization. Other peptide preparation methods tested, such as methanol-chloroform extraction with overnight trypsin digestion, often resulted in loss of hydrophobic peptides common to

MCS proteins. Peptides were resuspended in 1% formic acid + 1% acetonitrile at a concentration of 0.75 μg/μl prior to loading on instrument.

### Generation of MCS-PRM peptide library
A list of identified MCS proteins known to be expressed in human fibroblasts, derived from manually searching research publications and confirming localization via UniProt, was compiled and loaded into the Skyline targeted MS analysis platform[104,105]. Proteins were parsed for tryptic peptides between 6 and 30 amino acids in length, with only carbamidomethyl cysteine (C) modifications allowed, and to avoid peptides with multiple ion charge states. Transition ion settings were set to include 2 or 3 precursor charges, 1 or 2 ion charges, and y and b ions, with a library match tolerance of 0.5 $m/z$, instrument settings set to 50–1500 $m/z$ and a method match tolerance of 0.055 $m/z$, and full-scan settings with 15,000 resolving power at 200 $m/z$ for MS1 filtering and 30,000 resolving power at 200 $m/z$ for MS/MS filtering.

To experimentally build and validate the MCS-PRM peptide library, a subset of high abundance MCS and control peptides were first identified by analyzing uninfected whole-cell lysates by DDA-MS (see below). DDA-MS RAW files were uploaded directly into Skyline and searched in Proteome Discoverer (ThermoFisher Scientific) to generate a spectral library for ID'd peptides, which was also loaded into Skyline. Using the LC retention times (RT) for peptides identified by DDA-MS and the retention time predictor calculator in Skyline, a series of scheduled runs (8-minute RT windows) were performed to search for the bulk of MCS peptides, further improving the RT predictor by including additional ID'd peptides after each run. Initially searching for approximately 8 peptides per protein, we defined our final MCS-PRM library with ≥2 peptides per protein, reliably identified with a mass error (ppm) of ≤10 (primarily ≤ 5 ppm, see Supplementary Fig. 1). Three endogenous peptide controls were also included in the assay: histone (H2A1A & H2B1A) and tubulin (TBB5) peptides (see Supplementary Data 1). These controls were selected because they do not change abundance during infection, as evident during PRM and DDA MS analyses from this study (data available in the PRIDE repository, see above) and previous reports[4,17]. For experimental use, isolation lists included a maximum of 18 concurrent precursors at 5 or 8-minute RT windows, aiming for ≥10 MS/MS scans per ion peak.

### Parallel reaction monitoring (PRM) and data-dependent (DDA) MS data acquisition
All MS data was acquired via nano-liquid chromatography-mass spectrometry (LC-MS/MS) using a Dionex Ultimate 3000 nanoRSLC (ThermoFisher Scientific) and Thermo Q-Exactive HF orbitrap mass spectrometer. 1.5 μg (2 μl) of sample was loaded onto a 25 cm EASY-Spray HPLC column (ThermoFisher Scientific), and peptides were separated over a 60-minute gradient (2 μl/min flow rate) of 0-35% mobile phase A (0.1% formic acid) to B (0.1% formic acid in 97% acetonitrile). PRM methods consisted of targeted MS2 scans with a resolution of 120,000, AGC target 5e5, maximum inject time of 200 ms, isolation window of 1.2 $m/z$, and normalized collision energy of 27. Peptide isolation lists, generated in Skyline with the parameters described above, were pre-loaded into the instrument method. For DDA, a Top20 full MS to dd-MS2 scan method was used. Full MS scans had resolution of 60,000, AGC target 3e6, maximum inject time of 30 ms, and a scan range of 350−1500 $m/z$; dd-MS2 scans had a resolution of 15,000, AGC target of 1e5, maximum inject time of 42 ms, isolation window of 1.2 $m/z$, and normalized collision energy of 28.

### MCS-PRM analysis pipeline
MS RAW data files were loaded into and analyzed in Skyline[104,105]. In parallel, RAW files were searched by Proteome Discoverer 2.4 (ThermoFisher Scientific) to ID peptide spectra, and search results were loaded into Skyline as a spectral library to ensure peptide identification

for every experiment. In short, the Proteome Discoverer processing workflow used a FASTA file containing human protein sequences (downloaded from Uniprot in January 2021) and common contaminants, and the SEQUEST HT algorithm to analyze MS/MS spectra. SEQUEST was set to search for tryptic peptides with ≤2 missed cleavages, a precursor mass tolerance of 10 ppm, fragment mass tolerance of 0.02 Da, carbamidomethyl cysteine (C) modifications, dynamic deamidation of asparagine, dynamic oxidation of methionine, and dynamic loss of methione plus acetylation of the protein N terminus. Matched spectra were validated by the Percolator node.

Separate Skyline files were kept for each infection time course. For each infection, peptides were quality-controlled for those found across all infection timepoints and fragment ions were parsed for the 3-8 most intense and reproducible ions across samples (ppm ≤ 10, steady relative contributions to peptide peak area). Skyline peptide and transition settings are defined above. A single Excel file containing peptide intensities (not yet scaled or normalized to standard peptides) for each sample was exported for further processing. Specifically, each peptide value was normalized to the three control peptide intensities for each sample, and then scaled to the peptide abundance mean across timepoints per biological replicate. These values were used to score the coefficients of variation (CVs) for peptide quantifications across biological replicates (Supplementary Fig. 2C) and for ANOVA statistical analyses (Supplementary Fig. 2D). After normalization and scaling, all peptides per protein were averaged for each timepoint per replicate, and then averaged across replicates to get a single value per timepoint (as plotted in Fig. 1). The average peptide abundance standard deviation between biological replicates was calculated for protein values (see Supplementary Data 2). For heatmaps, protein abundance values were calculated as a ratio to the mock/uninfected timepoint.

### Proteomic comparisons of MCS-PRM data
For comparisons of MCS protein abundances to whole-proteome datasets in Supplementary Figs. 3-5 and Supplementary Data 3, marker proteins used for localization and function were curated from the Human Protein Atlas[106] and Gene Ontology Consortium[107,108], respectively. These proteins are listed and categorized in Supplementary Data 3A, and their abundance values for each infection is specified in Supplementary Data 3B-E. In total, comparisons include >5,200 proteins assigned to the localization and functional categories described below and in Supplementary Figs. 3-5.

For subcellular localization, proteins selected from the Human Protein Atlas exhibited "enhanced" confidence scores, with the exception of the peroxisome, lysosome, and lipid droplet, in which the cut-off was extended to "enhanced" and "supported" due to the small number of proteins (<5) in the "enhanced" category. "Enhanced" and "supported" confidence scores are the top two confidence categories in the Human Protein Atlas, based off microscopy analyses for reliable subcellular markers. The number of markers we used for each organelle was as follows: ER ($N = 48$), mitochondria ($N = 123$), peroxisome ($N = 15$), endosome ($N = 7$), lysosome ($N = 17$), Golgi ($N = 71$), plasma membrane ($N = 121$), lipid droplet ($N = 16$), cytosol ($N = 350$), nucleus ($N = 793$).

For functional categorization, gene ontology terms were searched using AmiGO[109], picking human proteins with similar functions to MCS proteins: cellular lipid metabolic process (GO:0044255; $N = 327$ proteins), intracellular calcium signaling and transport (GO:0035584, GO:0051924; $N = 52$ proteins), organelle biogenesis (peroxisome GO:0007031, mitochondrial fusion GO:0008053, organelle assembly GO:0070925, autophagosome GO:0000045; $N = 945$ proteins), vesicle transport (GO:0016192; $N = 995$ proteins), cellular signaling (apoptotic process GO:0006915, growth factor GO:0070848; $N = 995$ proteins), and mitochondrial organization (GO:000700; $N = 97$ proteins).

Proteins from each list were then searched in data-dependent (DDA) MS whole-proteome datasets of each infection, using both previously published studies and analyses from this work: Jean Beltran PM

et al., *Cell Systems* 2016 (HCMV), Lum K et al., *Cell Systems* 2018 (HSV-1), this study (Influenza A and HCoV-OC43). The DDA datasets from this study were acquired with the technical and experimental parameters described in the relevant Methods sections above. The DDA datasets from Jean Beltran et al. and Lum et al. were also acquired using the same instrumentation and similar detection parameters[4,110]. Each dataset was normalized to its specific control condition (e.g., Mock/uninfected) prior to comparison to MCS-PRM data to control for technical variation across experiments, so that only fold-change values were being compared. For each localization and abundance category, the median protein abundance value for all proteins assigned to that that category and detected by DDA-MS was calculated. MCS-PRM data was divided by these median values, with each MCS protein compared to its organelle localization or primary function. All of these values are listed in full in Supplementary Data 3B-E. Multi-localized and multi-functional proteins were tested for each known localization and function, as referenced in Supplementary Table 1 and depicted in Supplementary Figs. 3-5.

### Transfections for imaging, knockdowns, and overexpressions

To transfect MRC5 human fibroblast cells with high efficiency (% of cells expressing plasmid), low cytotoxicity, and baseline organelle dynamics (as determined by comparison to IF or staining with fluorescent dyes), we tested variables including cell density, transfection reagent, amount of plasmid DNA, cell passage number, and cell cycle progression. We found that MRC5 cells could not be properly transfected past passage 26, and cell stocks purchased from ATCC are already at passage 17 or more. In addition, cell density was key and varied for plasmid DNA versus siRNA transfections, and we found that passaging cells after transfection increased the number of cells expressing the plasmid by ~20%. This was important, as the number of cells expressing the plasmids reduced significantly across the duration of 5-day HCMV infections.

For all DNA transfections (imaging and overexpressions) in MRC5 cells, cells were plated in 6 cm dishes or 6-well plates and transfected when cells were approximately 75% confluent. Transfection mixes of plasmid DNA and XtremeGENE HP DNA transfection reagent (Sigma Aldrich) were incubated in OMEM for 20 minutes, using a 1:3 ratio of µg DNA to µl XtremeGENE. Transfection mixes were added to MRC5 cells in half-volumes of fresh OMEM and left to incubate for 5.5 hours. Cells were then rinsed in PBS, trypsinized and spun down, resuspended in complete DMEM, divided into new dishes (e.g., 3x2mL wells from a 6 cm dish, or 4×0.5 mL wells from a 2 mL well), and left to recover for 48 h, at which time they robustly expressed the plasmid DNA and had reached 90-100% confluency, as necessary for virus infection.

For all knockdowns (KDs) in this study, cells were grown to 65-70% confluency prior to KD. 80 picomoles of siRNA oligos were used per 2 mL well of cells (approximately 150,000 cells). siRNA was incubated in Lipofectamine RNAiMAX (ThermoFisher Scientific) and OMEM for 5 minutes, according to the manufacturer protocol, before being added to cells in fresh, complete DMEM. KD cells were left for 48 hours before further experiments, and were not passaged during siRNA transfections. For virus infections, fresh siRNA, RNAiMAX, and OMEM was re-added to the media after the inoculation period. KDs were confirmed by western blotting, IF, or targeted MS quantification.

DNA nucleofections into RPTE cells were performed using the Lonza Nucleofector 2b (AAB-1001) with the Lonza Amaxa Basic Nucleofector Kit for Primary Epithelial Cells (VAPI-1005) using the U-017 nucleofection protocol. RPTE cells were grown to 80% confluency in 10 cm plates, trypsinized with 0.25% Trypsin (Corning, 25-053-CI), and allowed to come off the plate. Trypsin was neutralized using soy trypsin inhibitor (Sigma, T6522-5X100MG) in PBS and the cells were centrifuged at 300 x g for 5 minutes. After this, 5 µg of plasmid was used to nucleofect approximately $1.5 \times 10^6$ cells per plasmid. From each nucleofection, $2.5 \times 10^5$ cells were seeded into 12-well plate wells and allowed to recover for 3 days.

### Antibodies, plasmids, and siRNA oligos used in this study

Host protein antibodies used: PTPIP51 (also known as RMDN3, 1:1000 for IF and PLA, 1:250 for WB, Sigma HPA009975); VAP-B (1:200 for IF and PLA, ProteinTech 66191-1-Ig); STING (1:400 for IF, Abcam Ab198950); ACBD5 (1:500 for IF, 1:150 for WB, Sigma HPA012145); PEX14 (1:500 for IF, Abcam ab183885); IRF3 phospho-S386 (1:1000 for WB, Abcam ab76493); TBK1 phospho-S172 (1:1000 for WB, Cell Signaling 5483 S); IRF3 (1:1000 for WB, Abcam ab68481); TBK1 (1:500 for WB, Cell Signaling 3504 S); PARP (1:500 for WB, Cell Signaling 9524); α-tubulin (1:5000 for WB, Sigma Aldrich T6199); GAPDH (1:4000 for WB, Cell Signaling D16H11-5174S); STING (Alexa Fluor 488 conjugate, 1:400 for IF, Abcam ab198950).

Viral protein antibodies used: IE1 (HCMV, 1:40 for IF, 1:100 for WB, gift from Dr. Thomas Shenk, Princeton University); pUL26 (HCMV, 1:100 for WB); pUL99 (HCMV, 1:40 for IF, gift from Dr. Thomas Shenk, Princeton University); ICP4 (HSV-1, 1:100 for IF, SantaCruz sc-69809); NP (Infl. A, 1:60 for IF, gift from Dr. Thomas Shenk, Princeton University); N (OC43, 1:100 for IF, 1:2000 for WB, Millipore Sigma MAB9013).

Other antibodies/stains: Alexa Fluor Plus IgG (H + L) highly cross-adsorbed secondary antibodies (1:2000 for IF, 1:10,000 for WB, ThermoFisher Scientific: Goat anti-Mouse 488 A11001, Goat anti-Mouse 568 A11019, Goat anti-Mouse 647 A32728, Goat anti-Mouse 800 A32730, Goat anti-Rabbit 488 A32731, Goat anti-Rabbit 568 A11011, Goat anti-Rabbit 647 A32733, Goat anti-Rabbit 680 A27042, Goat anti-Rabbit 800 A32735); DAPI (1:1000, ThermoFisher Scientific #62248).

All antibodies used are validated by the companies described above; specific papers are provided on the companies' websites. We validated the anti-PTPIP51 and anti-ACBD5 in knockdown backgrounds, using IF and Western blotting, prior to proximity ligation analyses. The antibodies against viral proteins (from the group of Thomas Shenk) have been previously validated by immunoaffinity purification of the targeted proteins.

Plasmids were either cloned in-house, purchased from Addgene, or received as gifts from the groups indicated below. Plasmids, with amount (ng of DNA) used for a 2 mL tissue culture dish (e.g., approximately 150,000 MRC5 cells upon transfection), were: eGFP-Sec61β (600 ng, gift from Dr. Tom Rapoport, Addgene plasmid #15108); mCh-Sec61β (800 ng, gift from Dr. Gia Voeltz, Addgene plasmid #49155); mito-BFP (600 ng, gift from Dr. Gia Voeltz, Addgene plasmid #49151); eGFP-VAP-B (100 ng, gift from Dr. Catherine Tomasetto, Addgene plasmid #104448); mito-ER synthetic tether (5AKAP(34-63)-9x-mRFP-9x-ER(ubc6), 500 ng, gift from Dr. Gyorgy Hajnoczky); eGFP-PTS1 (200 ng, gift from Dr. Michael Davidson, Addgene plasmid #54501); mCherry-PTS1 (300 ng, gift from Dr. Michael Davidson, Addgene plasmid #54520); eGFP-ACBD5 (250 ng, cloned in this study from human MRC5 cDNA); eGFP-OMP25 (300 ng, gift from Dr. Gia Voeltz, Addgene plasmid #141150). Expression was confirmed via confocal microscopy, and baseline expression levels were determined as visible fluorescence without disruption of organelle shape/dynamics or cellular viability.

siRNA oligos were designed in-house and purchased via Millipore Sigma (Custom siRNA), with dT[dT] 5' caps and no modifications. Two siRNAs targeting different sequences in the transcript were used in parallel for all experiments. siRNA oligo sequences were:

| | |
|---|---|
| **Control** (against GFP) | GGUGUGCUGUUUGGAGGUCTT |
| **PTPIP51 #1** | CCUUAGACCUUGCUGAGAUUU |
| **PTPIP51 #2** | GAAGCUAGAUGGUGGAUGAUU |
| **ACBD5 #1** | CCGTTAATGGTAAAGCTGAAA |
| **ACBD5 #2** | GCACAGTGGTTGGTGTATTTA |

## Virus titering

To measure virus production in control, KD, and OE conditions, media containing progeny virus was collected from cells at the end of one round of the virus replication cycle: 120 hours for HCMV, 24 hours for HSV-1, Infl. A, and HCoV-OC43. This was centrifuged to pellet cell debris (300 x g for 3 minutes), and supernatant was used to make serial dilutions (1 – 1:10,000) of produced virus. 96-well plates containing confluent MRC5 cell (HCMV, HSV-1, Infl. A) or LLC-MK2 cell (HCoV-OC43) monolayers were infected with the serial dilutions and fixed in 4% paraformaldehyde at 24 hpi (HCMV), 8 hpi (HSV-1), 8 hpi (Infl. A), or 12 hpi (HCoV-OC43). Plates were assayed for the immunofluorescent expression of viral proteins: IE1 for HCMV, ICP4 for HSV-1, NP for Infl. A, and N for HCoV-OC43. Infected cells were detected and quantified by a Perkin Elmer Operetta automated microscope. Dilutions in which approximately 30% of cells were infected in the control sample were used to calculate infectious units. Every titer experiment included at least 2 biological replicates and 2 technical replicates.

For all overexpressions (OEs), a plasmid with a fluorescent protein localized to the appropriate organelle was used as control (e.g., mito-BFP for mito-ER tether, eGFP-PTS1 for ACBD5 OE). For all KDs, a non-targeting or siGFP control KD was performed in parallel.

## Manipulation of herpesvirus temporal gene expression

For investigating the requirement of viral gene expression (as in Figs. S6), HCMV virions were UV-irradiated prior to infection, or the small molecule inhibitor phosphonoformic acid (PFA, also known as Foscarnet, for HCMV) was added to cell media following virus inoculation. For UV-irradiation, aliquots of purified virus (contained in microcentrifuge tubes) were left under a UV lamp in a sterile biosafety hood for 2 hours. An uninfected "Mock" condition was paired with UV-irradiated infection samples, same as for control samples. Confirmation of UV-irradiation included monitoring nearly complete inhibition of viral gene expression by targeted MS and Western blotting (Supplementary Fig. 6B, 6F). For PFA treatments, the drug was reconstituted in DMSO and then added to cells at a final concentration of 0.4 mg/mL. An uninfected, drug-treated control was used as the "Mock" condition for PFA infection samples. Confirmation of drug-induced suppression of late viral gene expression included monitoring viral genes from each temporal class (immediate early, delayed early, late) by targeted MS or Western blotting, and observing the expected decrease in protein levels compared to a wildtype infection (Supplementary Fig. 6B, S6F). Note that, in both conditions, some viral genes are still detected due to being carried in with wildtype virions. Cells were infected with the same MOI as the control/wildtype condition (MOI = 3 for HCMV).

## Sample preparation for live-cell and immunofluorescence microscopy

For live-cell microscopy experiments, cells were plated on 35 mm glass-bottom MatTek dishes that had been coated with human fibronectin (VWR, #354008), and 48 hours later they were imaged for an uninfected timepoint. Cells were then infected with HCMV immediately after imaging, and the same dish of cells was imaged every 24 hours across the duration of HCMV infection. For immunofluorescent (IF) experiments, cells were plated on fibronectin-coated glass coverslips. At each timepoint indicated, cells were rinsed with warm PBS, fixed in 4% paraformaldehyde + 0.25% glutaraldehyde for 15 minutes, rinsed with PBS, and permeabilized with cold methanol at -20 °C for 15 minutes. At room temperature, samples were blocked in 5% goat serum + 5% human serum in PBS + 0.1% TritonX100, stained with primary antibodies in block for 2 hours, and stained with secondary antibodies for 45 minutes. Coverslips were mounted on glass slides with ProLong Diamond antifade (ThermoFisher Scientific, P36970).

## Image acquisition and analysis

A Nikon Ti-E inverted fluorescence confocal microscope (Nikon Instruments, Melville, NY) equipped with a Yokogawa spinning disc (CSU-21), digital CMOS camera (Hamamatsu ORCA-Flash TuCam), and precision microscope stage (Piezo) was used for most imaging experiments in this study. Where specified, a Nikon Ti-E2 microscope equipped with a CSU-W1 SoRa super-resolution module was used. Live-cell experiments were acquired with 5- or 2-second intervals over 2-3 minutes using a Nikon 100X or 60X Plan Apo objective, and cells were kept in a heated and humidified chamber during imaging. For immunofluorescent experiments, z-stacks were acquired with 0.1-3μm steps throughout the cell depth. Where applicable, images were processed with Nikon Denoise.ai and 3D deconvolution in NIS-Elements AR, or processed with ImageJ (National Institutes of Health) and displayed with a background subtraction (rolling ball radius = 200 pixels) applied to each channel. All scale bars correspond to 10μm in length.

ER-mitochondria colocalization was performed with the ImageJ Coloc2 plug-in on z-stack images. ER-mitochondria line-scans were done by drawing a straight line along the mitochondrial length (Fig. 2) or across ER-mitochondria junctions (Figs. 2–3) and calculating the fluorescence intensity profile for each channel with ImageJ ("Plot Profile" command). Intensities were scaled from 0 to 1 for each channel and smoothed to the 2nd order in Prism 8. Changes to ER-mitochondria contact structure, temporal stability, and PTPIP51 re-localization were manually scored for ~20 mitochondria per cell that could be resolved as individual mitochondria in the xy plane ($N > 34$ cells per timepoint for ER-mitochondria structure, N > 25 for temporal stability, $N > 16$ for PTPIP51). For live-cell analyses, mitochondria that moved out of focus were not included. For the mitochondria fragmentation analysis, mitochondria from a single z-stack plane were manually thresholded and surface area was measured with the "Wand (tracing) tool" in ImageJ for >15 mitochondria per cell ($N \geq 20$ cells per timepoint). For the ER density analysis, three 7×7 μm ROIs distributed throughout the cell were thresholded and the fraction of black-to-white pixels was recorded and averaged for each cell.

For the IE1 expression analysis upon mito-RFP-ER tethering, individual nuclei stained for endogenous IE1 at 24 hpi were first filtered for detectable IE1 expression, using a standard fluorescent intensity cutoff across all conditions. IE1-positive nuclei were then counted as a proportion to all nuclei (DAPI stained) by a Perkin Elmer Operetta automated microscope. IE1 intensity per individual nuclei (not including IE1-negative nuclei) was then quantified and plotted. This analysis included two biological and two technical replicates per condition.

Peroxisome motility was assessed by the Mosaic 2D/3D Particle Tracker in ImageJ, with the following parameters: radius of 8, 0.001 cutoff, 0.500 per/abs, link range 2, displacement 10, Brownian motion. Only peroxisomes that were tracked in a single plane for more than 20 frames (100 seconds) were used for further analysis. Peroxisome trajectories were loaded into Python, and 2D histograms were generated using the Seaborn data visualization library. The analyses of ER-associated peroxisome dynamics and enlarged peroxisomes on dense ER were done manually by visually tracking 20 peroxisomes (dynamics) or all enlarged peroxisomes in the field of view for the movie duration. Peroxisome size and numbers were quantified with the 3D Objects Counter ImageJ plugin (Bolte and Cordelières, 2006), with a minimum size defined as 10 consecutive voxels (at a resolution of 0.065 μm/pixel). For ACBD5 co-localization, peroxisomes were divided by surface area distribution (small = <0.9μm², medium = 0.9-3.1μm², and large = >3.1μm²), and the 3D ROI Manager was used to quantify ACBD5 intensity within each peroxisome. Three-dimensional movies were generated using the volume viewer and movie maker in NIS Elements (Nikon) software.

## Proximity Ligation Assay (PLA)

MRC5 cells in glass-bottom scope dishes were transfected with organelle-targeted fluorescent plasmids, fixed, and stained with optimized primary antibodies as described above. A DuoLink PLA In Situ Red Kit (Sigma Aldrich) was used for PLA according to the manufacturer protocol. In short, cells were stained with PLA probes for 1 hour at 37 °C, incubated in ligation solution for 30 minutes at 37 °C, and underwent amplification for 100 minutes at 37 °C. Great care was taken to keep cells in the dark and prevent samples from drying out, as volumes of only 50 μl were used and drying increases background fluorescence. Scope dishes were imaged immediately after amplification, collecting z-stacks throughout the cell depth. PLA signal was quantified by hand-counting the number of puncta (peroxisomes) or, in the case of PTPIP51-VAP-B PLA, by total cellular intensity, as there were too many overlapping puncta to quantify in late infection timepoints (N > 38 cells per timepoint).

## Western blots

Cell lysates were collected via trypsinization, pelleted at 10,000 rpm in a benchtop centrifuge for 3 minutes, and lysed in 5% SDS lysis buffer (same as for MCS-PRM analysis) with one round of sonication and heating at 70 °C. Lysates were diluted in Laemmli Buffer supplemented with 5% beta-mercaptoethanol (BME), heated at 105 C for 10 minutes, and loaded into handmade 10% polyacrylamide SDS-PAGE gels. Proteins were transferred to a PVDF membrane, blocked in 5% NF milk in tris-buffered saline (TBS), stained with primary antibody in block + 0.2% Tween20 overnight at 4 °C, and stained with secondary antibody in block + 0.2% Tween20 + 0.01% SDS for 45 minutes at room temperature. Some blots were stripped and re-stained for additional proteins by using REBLOT plus (Millipore Sigma) according to the manufacturer protocol. All blots were imaged with an Odyssey CLX system (Li-Cor), and densitometry was performed in ImageJ.

## TUNEL assay

Each TUNEL experiment included three biological replicates for each condition and the following controls: 1) non-transfected wildtype cells for baseline levels of apoptotic cells, 2) wildtype cells treated with DNase (10 minutes) for a positive control, and 3) wildtype cells for a negative control (not treated with TUNEL enzyme). A Roche In Situ Cell Death Detection Kit (Sigma Aldrich) was used to detect apoptotic cells in 96-well plates. In short, cells were fixed in 0.1% TritonX + 0.1% sodium citrate in PBS for 2 minutes at 4 °C. All wells were washed in PBS, and then positive control wells were treated with 10 units of DNAse I recombinant + 1 mg/mL BSA + 1 M Tris HCl pH 7.5 in water for 10 minutes at room temperature. All wells were washed in PBS again, and then incubated for 1 hour at 37 °C in TUNEL mix (enzyme solution + label solution). Cells were stained with DAPI and imaged with a Perkin Elmer Operetta automated microscope, scoring apoptotic cells by those positive for TUNEL staining.

## Interferon secretion assay

Secreted interferons were quantified using the LEGENDplex™ Human Anti-Virus Response Panel (BioLegend, #740390) according to the manufacturer's protocols, in technical duplicate. In short, all media from a 2 mL well of cells (genetic conditions and infection timepoints specified in Fig. 4l) was collected and immediately frozen at -80 °C. Immediately prior to sample analysis, the aliquots were thawed and 25 μl was used for binding to interferon-specific fluorescent capture beads, compared to a Human Anti-virus Response Panel Standard Curve, and analyzed by a BD LSR II flow cytometer. Beads were separated by size and fluorescence (reporter channels PE, 575 nm, and APC, 660 nm) into interferon-specific gates, and fluorescence intensity per bead was used to calculate abundance (in pg/mL) by comparison to the standard curve.

## STING and TBK-1 Loss-of-Function

For assessment of STING and TBK-1 contribution to mitochondria-ER tethering phenotypes (as in Fig. 4 and Supplementary Fig. 11), fibroblast cells were treated with STING and TBK-1 inhibitors in media for 2 hours prior to infection. Each drug (STING inhibitor H-151[93], TBK-1 inhibitor GSK8612[94]) was reconstituted in DMSO, and added to cells using two concentrations (0.05 and 0.1 μM), as reported in previous studies. After inoculation with HCMV, media containing fresh drug was added back to the cells, and infection supernatants were collected for titering at 120 hpi. A DMSO-only control was used for each genetic condition (mito-BFP control and mito-RFP-ER).

## STS-induced mitochondrial fragmentation

MRC5 fibroblasts, transfected and plated on scope dishes as described above for image analysis, were treated with 20 μM z-VAD-fmk caspase inhibitor (Cayman Chemicals, #14463) for 3 hours to prevent drug-induced apoptosis. Then, cells were treated with 15 nM staurosporine (STS, Cayman Chemicals) in complete media for 2 hours, prior to fixing and staining for PTPIP51, as described above. A control DMSO treatment was performed in parallel.

## Statistics and Reproducibility

All PRM data was loaded into and analyzed in Skyline (MacCoss Lab, University of Washington).

All heatmaps, graphs, and statistical analyses were done in GraphPad Prism 8, the only exception being the 2D histograms of peroxisome motility (supplemental figures), which were assembled in Python. Throughout this study, * signifies $p \leq 0.05$, ** $p \leq 0.01$, *** $p \leq 0.001$, and **** $p \leq 0.0001$ as determined by the significance tests indicated in figured legends. For all MS, molecular virology, and microscopy data in this manuscript, the quantification workflows, software, replicates (N values), results, and graphical display keys can be found in the figure legends and/or experiment-specific detailed descriptions are included in the Method Details sections above. All representative microscopy images included in main and supplementary figures are representative of two or more (>20) independent experiments that obtained similar results, imaging ≥10 cells per experiment per timepoint/condition, and phenotype quantifications are included for nearly all imaging experiments.

## Reporting summary

Further information on research design is available in the Nature Research Reporting Summary linked to this article.

# Data availability

All MCS-PRM MS data generated in this study has been deposited to Panorama Public[111] [https://panoramaweb.org/mcsPRMviruses.url]. The RAW data and a compiled MCS peptide library can also be found at ProteomeXchange with the dataset identifier PXD023761. Source data for microscopy, graphical representations, western blotting, and genetic perturbations included in all main figures and supplementary information are provided with this paper (see Source Data file). Source data are provided with this paper.

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

## Acknowledgements

We are grateful to Dr. Gary Laevsky (Princeton Confocal Imaging Facility) for training in microscopy, members of the Cristea Lab, in particular J.L. Justice and J.D. Federspiel for helpful insights, and funding from the NIH (GM114141), Stand Up To Cancer Convergence 3.1416, and a Mallinckrodt Scholar Award to I.M.C., an NIAID Predoctoral Fellowship (F31AI147637) and Princeton Centennial Award to K.C.C., and the NIGMS (T32GM007388).

## Author contributions

K.C.C. and I.M.C. designed research; K.C.C., E.T., and J.N. performed experiments; K.C.C., E.T., J.N., S.T., and I.M.C. analyzed data; and K.C.C., E.T., and I.M.C. wrote the manuscript.

## Competing interests

The authors declare no competing interests.
