## [Peer Review File · Nature Communications]

Restructured membrane contacts rewire organelles for human cytomegalovirus infectionThis manuscript has been previously reviewed at another journal that is not operating a transparent peer review scheme. This document only contains reviewer comments and rebuttal letters for versions considered at *Nature Communications*.

REVIEWER COMMENTS

Reviewer #1 (Remarks to the Author):

My concerns about this study remain the same. The central claim is that membrane contact site (MCS) protein abundance is specifically regulated by the viruses to optimize viral propagation. If true, this is a significant conceptual advance. However, I continue to feel the supporting evidence, while compelling, is not strong enough for this study to be appropriate for a high-impact journal. Here are my major concerns, numbered in the same order as my previous review.

1. There is already evidence that viruses substantially alter cellular metabolism in many ways (PMID: 31319842). The question here is whether changes in MCS protein abundance is specific or if there are similar changes in non-MCS proteins involved in related metabolic pathways. In my previous review I that asked that the changes in MCS protein abundance be compared to changes in the abundance of proteins with similar functions. This has been done (Fig. S3B), but it is not clear what groups of proteins are being compared or how to interpret the results. For example, the abundance of some MCS proteins is compared to that of “calcium transfer” proteins. What are these proteins, given that most of the major players in Ca²⁺ homeostasis are listed as MCS proteins? The MCS proteins involved in lipid metabolism also need to be compared to other lipid metabolic enzymes (not just those involved in lipid transport) and these need to be listed. In general, the larger question is whether changes in MCS protein abundance following viral infection occurs because MCS proteins are specifically being regulated or because whole metabolic pathways are being altered and these pathways include MCS proteins.

2-4. An additional way to make a stronger case that MCS protein abundance is specifically regulated by viruses would be to get some mechanistic insight into how regulations occur. Obtaining mechanistic insight requires more than correlating changes in protein MCS abundance with changes in cell structure, which is what is shown here. The new evidence added to address this concern is well done but does not demonstrate that virus specifically regulate MCS protein abundance.

5. I am still a bit perplexed by how MCS proteins were chosen. In the rebuttal letter, the authors say they did not include every protein ever reported to be at MCSs because they want to make a tool to study “MCS regulation.” I am not understand why including all known or suspected MCS proteins would reduce the usefulness of the tool. As I mentioned in my previous review, some proteins have been included in the list that are only slightly or transiently enriched at contact sites while others have been excluded. Why not include them all or make a list of proteins that are highly enriched at MCSs and those that are moderately/transiently enriched?

Reviewer #2 (Remarks to the Author):

In this manuscript, Cook et al describe the implementation of a targeted MS method based

on parallel reaction monitoring to accurately quantify membrane-contact-sites (MCS) associated proteins. This methodological approach allows the authors to quantify the relative abundance of those proteins over the course of diverse viral infections. The authors use a targeted proteomics approach to generate hypotheses for testing using microscopy, proximity ligation assays and genetic tools, to aid a deeper understanding of targeted MCSs interactions for the different viruses. This study primarily focuses on mitochondria-ER and peroxisome-ER interactions upon HCMV infection. Using live microscopy, the authors discovered that the mitochondria-ER interactions are rewired in late HCMV infection forming encapsulating structures formed from mitochondria that are driven by PTPIP51 tethered to VAP-B. They demonstrate that this interaction is required for optimal viral replication. Likewise, they find that ACBD5-mediated peroxisome-ER contacts are also key for HCMV, HSV-1, and Infl. A replication. The authors provide a valuable resource for virologists interested in how different enveloped viruses can hijack MCSs.

This is the second time I have reviewed this manuscript and I thank the authors for answering the copious queries I had pertaining to the first version. The manuscript is much more streamlined, easier to navigate and now more suited to be published in Nature Communications. I particularly appreciate the addition of organelle markers to facilitate determining MSC protein abundance changes over total organelle specific protein abundances not associated with MCSs.

Minor comments

1. In the methods section, there is now an additional paragraph 'Proteomic comparisons of MCS-PRM data'.

There are a few additions that would make this section more understandable to the reader.

a. How many markers are used– to save the reader trawling the supplementary tables

b. An explanation to the uninitiated what is meant by enhanced and supported scores from HPA

c. If DDA data was used from previous studies to the one described here, the authors should clearly state that the conditions used in these additional studies were the same as used here. If this was not the case, the authors should state the justification for their incorporation here.

d. I'm afraid I'm a bit confused by figure S3A. I would clearly like to see the changes in abundance for the MCS protein with hpi/ virus and the organelle markers plotted separately, so it is clear if there are any organelle wide abundance changes upon viral infection. If I understand correctly - it is not very clearly stated in the methods section – for each protein and each virus the PRM data are displayed in the left of the panel for each virus in S3A as PRM data and then PRM corrected for 'organelle median'. The response of each organelle marker should be shown in a heatmap too and some indications of the spread of data around the median. At the moment all I can see in table S7 is a pre-calculated organelle median – the same is true for the functional median.

e. The data are qualitatively displayed, in so much there is no attempt to compare and results and in some cases there are some discrepancies between the two methods of calculating abundance change of MCS proteins along the infection timecourse. A statement that the trends are largely the same, but in some cases there are notable discrepancies for example, ORF8 Infl. A

The authors have gone some way to allay my concerns about overstating organelle localizations of MCS proteins when they could be present in more than one contact site and indeed, elsewhere in the cell.

I still think the authors should be more upfront about the limitations of their approach, however powerful, to manage the expectations of the researchers who use their PRM assays, along the lines of... 'our approach gives indications of which MCSs are being modulated upon viral infections, but as some MCS proteins will be involved in multiple different tethering events, and may have alternate functions elsewhere in the cell, additional validatory experiments such as microscopy and proximity ligation assays may be needed to support the data provided by PRM assays'.

Reviewer #3 (Remarks to the Author):

Cook et al. explore the role of membrane contact sites (MCS) and define mitochondrial-endoplasmic reticulum encapsulation structures (MENCs) in the context of viral infection, using human herpesviruses, coronavirus, influenza A. This is an intriguing and well-carried out study with interesting and novel results that lay the foundation for new inquiries into MCSs and their role in viral infection. This study is powered by the innovation of mass spectrometry approaches to detect all MCSs with functional specificity and quantify changes in their components induced by viral infection. Major findings from this study include:

- HCMV induces a global increase in MCSs while other infections have a much less dramatic effect. By comparison, HSV-1 has little effect on MCSs and Influenza A and CoV-OC43 decreases MCSs.
- HCMV infection rewires mitochondrial-ER connections into encapsulated structures—MENCs
- MENCs are enriched for PTPIP51 and this contributes to HCMV replication, but loss of PTPIP51 does not alter MENC structure.
- ER-mitochondrial tethering enhances STING-TBK1-IRF3 and HCMV reduces these contacts
- ER-peroxisome contacts are proviral

Issues for authors to address:

1. In the paragraph beginning on line 337, the authors state that they tested the anti-viral capacity (line 338) of increased ER-mito tethering. They show that tethering decreases HSV1, HCMV and InflA replication. The conclusion is that this result "suggests" that this is due to the increase in STING activation (line 345). This may be true, but the experiments performed provide only correlation and do not definitively define causation. Tethering could result in decreased dynamics of organelle trafficking or signaling or serve to stimulate other antiviral defenses not yet investigated. To firmly establish that the main consequence of tethering for infection is indeed increased STING signaling, can the defect in replication be fully rescued by STING knockdown in cells where tethering has been induced? Without this demonstration, tethering may restrict virus replication due to upregulation of STING, but other possibilities cannot be excluded. Then on line 524-526, it is stated: "we show that this anti-viral effect is due to enhanced activation of the STING-TBK1-IRF3 immune axis, preventing establishment of infection (Fig. 4)." But it really has not been shown to be due to only correlated with.

2. Figure 4 F and I both use heat maps. In F, dark purple indicates decreased abundance, whereas in panel I it indicates increased abundance. It would be helpful if all heat maps had the same colors indicating directionality especially since colors are coordinated throughout the manuscript.

3. References are still need some curation so that the discussion of the results and interpretation in the greater context of the field is not so cursory. Two examples are given, but there are many places where the discussion could be deepened by providing more robust discussion to provide context.

-Example 1: Ref 50 is cited to support the statement that HCMV inhibits cell signaling by internalizing and degrading EGFR (line 188). Ref50 supports this statement to a point showing that HCMV reduces cell surface and total EGFR, but more detailed work has shown that EGFR is targeted for degradation and that it and downstream EGFR signaling is disrupted. PMID: 27218650 PMID: 30089695. Further, the MCSs described could be important to aberrant downstream signaling described for HCMV PMID: 32493823 PMID: 31725811, activation of innate pathways PMID: 29021395, entry into latency PMID: 27974567, as well as trafficking virus capsids PMID: 32723814

-Example 2: Ref 69 and 70 are used to support the statement that HCMV inhibits STING (line 321). 69 is a paper on US9 and 70 is a review. There are a number of examples of HCMV gene products impacting cGAS/STING beyond US9 and that predate US9. PMID: 28132838 PMID: 29018427 PMID: 32466380. The regulation of cGAS/STING and its pathway in HCMV infection is very complicated and the cursory treatment of it in the discussion of the results does not do justice to how these findings contribute to the greater context of cGAS/STING. This is particularly true since STING has recently been shown to facilitate delivery of the capsid to the nucleus PMID: 34385328.

Reviewer #4 (Remarks to the Author):

The authors still failed to address my original concerns that there was no causative relationship provided between the virus titer and the STING immune signalling. The STING relocation was well demonstrated (e.g., in Fig. 4G), I agree, but it does not necessarily mean that the STING signalling suppressed the virus titer and increased the cytokine secretion.

As I suggested, the experiments should be performed with STING inhibitor or in STING-KO cells. Ideally, the experiments may also be performed with TBK1 inhibitor or in TBK1-KO cells.

We thank the editor and the reviewers for their careful assessment of our manuscript and for the insightful comments and recommendations that they made to further improve the study. We are also grateful for their enthusiasm for the work. We have now performed all of the suggested additional experiments and made text changes to the manuscript in order to address all comments. Altogether, these data include seven new panels to main figures (new Influenza A data in Figure 1C-D, VAP-B analysis in Figure 3C-G, additional PTPIP51 imaging in Figure 3E), sixteen new panels to supplemental figures (new Figures S3-6, Figure S8C-D, Figure S11G-I), and new data to supplementary Tables S5 and S7. Specific experimental and analytical advances in this revised manuscript include: **1)** Expanding our comparisons of MCS protein abundances defined by MCS-PRM to that of organelle or functional markers monitored by whole-cell proteomics, including analysis of over 5,000 human proteins from related functional pathways during each infection included in our study; **2)** Assessment of STING and TBK1 loss-of-function and contribution to the anti-viral effect of ER-mitochondria tethering during virus infection; **3)** Investigation of the requirement of HCMV viral gene expression for modulating MCS protein abundances; **4)** Expansion of the mechanism of PTPIP51 recruitment to mitochondria-ER encapsulations, including temporally monitoring the localization of its ER-resident tethering partner VAP-B across HCMV infection time. Detailed responses are given in the point-by-point responses below, which are marked with “>”. A manuscript with highlighted changes to the text is provided separately.

REVIEWER COMMENTS

Reviewer #1 (Remarks to the Author):

My concerns about this study remain the same. The central claim is that membrane contact site (MCS) protein abundance is specifically regulated by the viruses to optimize viral propagation. If true, this is a significant conceptual advance. However, I continue to feel the supporting evidence, while compelling, is not strong enough for this study to be appropriate for a high-impact journal. Here are my major concerns, numbered in the same order as my previous review.

1. There is already evidence that viruses substantially alter cellular metabolism in many ways (PMID: 31319842). The question here is whether changes in MCS protein abundance is specific or if there are similar changes in non-MCS proteins involved in related metabolic pathways. In my previous review I that asked that the changes in MCS protein abundance be compared to changes in the abundance of proteins with similar functions. This has been done (Fig. S3B), but it is not clear what groups of proteins are being compared or how to interpret the results. For example, the abundance of some MCS proteins is compared to that of “calcium transfer” proteins. What are these proteins, given that most of the major players in Ca²⁺ homeostasis are listed as MCS proteins? The MCS proteins involved in lipid metabolism also need to be compared to other lipid metabolic enzymes (not just those involved in lipid transport) and these need to be listed. In general, the larger question is whether changes in MCS protein abundance following viral infection occurs because MCS proteins are specifically being regulated or because whole metabolic pathways are being altered and these pathways include MCS proteins.

- To address this reviewer’s concern, we have further compared the MCS-PRM data to other cellular proteins that perform similar functions for every major MCS function included in our assay. Specifically, these comparisons now involve over 5,000 proteins derived from eleven gene ontology categories, and assigned to six major groups: lipid metabolic process (GO:0044255), intracellular calcium signaling and transport (GO:0035584, GO:0051924), organelle biogenesis (peroxisome GO: 0007031, mitochondrial fusion GO:0008053, organelle assembly GO:0070925, autophagosome GO:0000045), vesicle transport (GO:0016192), cellular signaling (apoptotic process GO:0006915, growth factor GO:0070848), and mitochondrial organization (GO:000700). Each of these proteins are organized by category and listed in Table S7A, including their UniProt accession numbers (as was included for the smaller list during the last submission). Every abundance value from the DDA-MS data (*i.e.*, protein abundances from whole

proteome analyses) for each of these proteins, from each infection, is given in Tables S7B-E, and again clearly labeled by functional category. We have further expanded the Methods and the figure legends to clarify that Table S7 lists out every protein used in these comparisons.

- We find that, in each functional category during each infection, the trends of the changes in MCS protein abundances are more pronounced than other proteins with similar function (Fig. S4). The enrichment trends are either profound or subtle, depending on the type of infection. For example, MCS proteins across functional categories in HCMV infection are highly enriched compared to non-MCS protein abundance changes, while HSV-1 retains mild changes throughout, in agreement with our original MCS-PRM analysis. To ensure that the scope of these comparisons are clear, we have divided previous Supplemental Figure S3 into three new figures: S3 (comparison of MCS-PRM to organelle/localization markers from DDA-MS data), S4 (comparison of MCS-PRM to functional markers from DDA-MS data), and S5 (revised heatmaps depicting protein abundances prior to and after comparisons). For each proteomic comparison, we have included additional line graphs (Fig. S3B-C, S4B-C), sorted by each localization or functional category, that illustrate the range of abundance changes from both the DDA-MS and MCS-PRM data.
- We have also edited the results section to include a more detailed description of the trends observed for MCS proteins compared to organelle and functional proteomes (pages 8-9). We indicate that, overall, changes in MCS protein abundances were enriched compared to non-MCS proteins of similar localization and functions, suggesting that these protein abundances are selectively altered over proteins of shared subcellular compartments or functional pathways. The abundance trends of MCS proteins had distinct magnitude and/or temporality compared to organelle and functional marker proteins, especially for HCMV and HCoV-OC43 (**Fig. S3B-C, S4B-C**). For example, during HCMV infection, Golgi MCS proteins are elevated throughout infection while the overall Golgi proteome decreased in abundance. As an example of distinct temporality, peroxisomal MCS proteins increased in abundance 72 hours prior to the observed global increase in the peroxisomal proteome (**Fig. S3C**). In contrast, select MCS proteins (e.g., the ER-resident VAP proteins during HCMV infection) appear to change in sync with proteins of related localization, with abundance trends being damped down after comparison to organelle marker proteins (**Fig. S5A**). For the functional comparisons, nearly all changes in MCS proteins were enriched above proteins of similar functional pathways for each infection examined (**Fig. S5B**).

2-4. An additional way to make a stronger case that MCS protein abundance is specifically regulated by viruses would be to get some mechanistic insight into how regulations occur. Obtaining mechanistic insight requires more than correlating changes in protein MCS abundance with changes in cell structure, which is what is shown here. The new evidence added to address this concern is well done but does not demonstrate that virus specifically regulate MCS protein abundance.

- The reviewer brings up an interesting point that we have considered, *i.e.*, are the alterations in MCS protein abundances driven by the viruses or a general host cell response to infection? In our paper, for the four viruses that we have tested, we found unique changes to MCS protein levels in the same cell type over similar time scales, where the only variable changing was the type of virus used. This suggests virus-type specific changes to MCS protein abundance rather than a general host response to infection. We also demonstrated that several of these MCS proteins (e.g., PTPIP51, ACBD5) are necessary for supporting virus production. However, to further address the reviewer's comment, we have now assessed the necessity of viral gene expression for modulating MCS protein abundances during HCMV infection. We used several approaches, which are included in the new Figure S6, presented in the Results section (pages 10-11), included in the Methods section, and summarized here:

- First, we used UV-irradiation to deactivate virus genomes, thus preventing viral gene expression. We confirmed the expected inhibition of viral gene expression by quantitatively monitoring viral proteins from each temporal expression class (immediate early, delayed early, late; Figure S6B). UV-irradiated HCMV lacks the expression of viral gene products needed to inhibit cell cycle progression, which leads to over-confluent cell cultures at later infection timepoints. Therefore, we focused the UV experiment on early infection (24 hpi). Using our MCS-PRM assay to compare UV-irradiated to wildtype virus infections, we find that virus gene expression is required for HCMV-driven alterations to MCS protein abundances (Figure S6C-D). Specifically, the wildtype trends in abundance are reversed, with relative decreases observed for the majority of MCS proteins in UV compared to wildtype HCMV infections. This demonstrates that HCMV gene products are required for the increase in MCS proteins that we have characterized with MCS-PRM.
 - Next, we used the established clinical drug phosphonoformic acid (PFA, also known as Foscarnet) to inhibit the viral polymerase and thus prevent expression of late viral gene classes (Fairley et al., 2002; Jean Beltran et al., 2018; Zahn et al., 2011). We performed this treatment throughout the virus replication cycle, in parallel with a repeat of the control infection, and analyzed each timepoint using our MCS-PRM assay. We find that late viral genes are required for altering the abundances of specific subsets of MCS proteins (Figure S6C-D). For example, PFA treatment prevents the increase in MCS proteins at ER contacts with other endocytic organelles (Golgi, plasma membrane, endosomes, lysosomes), while the majority of other trends remain the same as in untreated infected cells. This indicates that late viral gene products are needed for increasing the abundance of these specific ER MCS proteins. This may be connected to the functions of these proteins in lipid and calcium transfer (e.g., ORP5/8, CERT, STIM1, STIMATE, VAP-A/B), processes known to be modulated by late viral genes and involved in the generation of the viral assembly compartment late (72-120 hpi) during infection (Alwine, 2012).
 - To further validate these changes, we repeated the experiment and analyzed by Western blot, focusing now on the two proteins that we have investigated functionally in our paper: PTPIP51 and ACBD5. This orthogonal analysis again showed that their infection-driven increases in abundance are dependent on viral gene expression, as UV irradiation prevented the abundance changes. Additionally, PFA treatment has little to no effect on their upregulation, suggesting that immediate early or early viral genes are involved in the upregulation of ACBD5 and PTPIP51.
- We also further expanded our investigation of PTPIP51 at ER-mitochondria contact sites, defining the temporality of PTPIP51-VAP-B interactions induced by HCMV infection. Specifically, in both live and fixed cells we have now monitored the localization of VAP-B, the ER-resident tethering partner of PTPIP51 that is known to recruit PTPIP51 from the cytosol or mitochondrial outer membrane to ER contact sites. We now demonstrate that VAP-B accumulates at remodeled ER-mitochondria junctions by 48 hpi, approximately 24 hours prior to PTPIP51 (Fig. 3C-F). VAP-B localization at mitochondria-ER encapsulations (MENCs) is highly stable across both time and space, indicating its participation in stable membrane contacts between these organelles. PTPIP51 tethering interactions with VAP-B turn on at 72 hpi (Fig. 3H), which corresponds with the timing of PTPIP51 accumulation at MENCs (Fig. 3E). Given that PTPIP51 KD reduces virus titers by nearly 100-fold, our findings point to a mechanism whereby HCMV infection elevates the protein levels of VAP-B and PTPIP51 and triggers their accumulation at remodeled ER-mitochondria junctions late in infection to support virus production. This occurs by first forming MENCs, which recruit VAP-B and is followed by PTPIP51 recruitment, participating in stable ER-mitochondria tethering necessary for virus production. These results are described in the text on page 16.

5. I am still a bit perplexed by how MCS proteins were chosen. In the rebuttal letter, the authors say they did not include every protein ever reported to be at MCSs because they want to make a tool to study “MCS regulation.” I am not understand why including all known or suspected MCS proteins would reduce the usefulness of the tool. As I mentioned in my previous review, some proteins have been included in the list that are only slightly or transiently enriched at contact sites while others have been excluded. Why not include them all or make a list of proteins that are highly enriched at MCSs and those that are moderately/transiently enriched?

- We understand that our wording was not explicit enough. To clarify, MCS proteins were chosen to be included in our assay based on having reported functions linked to a specific MCS localization. A report of a localization of a protein at an inter-organelle junction was not deemed sufficient for including that protein in our MCS-PRM assay, without additional knowledge of the specific contribution of that protein to the formation of the organelle contact or a downstream function of the MCS. Several proteins that are only transiently enriched at MCSs (e.g., DRP1, DNM2, INF2 and NPC1) are included in our assay, given the defined functions of these proteins dependent on their MCS enrichment (e.g., ER-mediated mitochondrial fission and ER-endosome cholesterol transfer, respectively). For the MCS proteins included in MCS-PRM, Table S1 lists their defined tethering partners, known MCS localizations, contact-dependent functions, and citations for their characterized biological mechanisms. We have now clarified this in the legend for Supplemental Table S1 (Supplementary Information). We also went through the manuscript and ensured that our focus on functionally-characterized MCS proteins is clearly stated in all sections.
- We strive to provide a timely assay that captures MCS biology, and hence since the first submission of our manuscript we have expanded our assay by adding nearly 30 MCS proteins that were functionally characterized, optimizing their detection parameters for inclusion in MCS-PRM. We have now monitored all of these during all four infections included in our study – see updated Figure 1C-D, Table S3-6, including a third replicate of Influenza A infection added during this second revision cycle. We plan to continually expand the MCS-PRM library as the organelle contact field advances.

Reviewer #2 (Remarks to the Author):

In this manuscript, Cook et al describe the implementation of a targeted MS method based on parallel reaction monitoring to accurately quantify membrane-contact-sites (MCS) associated proteins. This methodological approach allows the authors to quantify the relative abundance of those proteins over the course of diverse viral infections. The authors use a targeted proteomics approach to generate hypotheses for testing using microscopy, proximity ligation assays and genetic tools, to aid a deeper understanding of targeted MCSs interactions for the different viruses. This study primarily focuses on mitochondria-ER and peroxisome-ER interactions upon HCMV infection. Using live microscopy, the authors discovered that the mitochondria-ER interactions are rewired in late HCMV infection forming encapsulating structures formed from mitochondria that are driven by PTPIP51 tethered to VAP-B. They demonstrate that this interaction is required for optimal viral replication. Likewise, they find that ACBD5-mediated peroxisome-ER contacts are also key for HCMV, HSV-1, and Infl. A replication. The authors provide a valuable resource for virologists interested in how different enveloped viruses can hijack MCSs.

This is the second time I have reviewed this manuscript and I thank the authors for answering the copious queries I had pertaining to the first version. The manuscript is much more streamlined, easier to navigate and now more suited to be published in Nature Communications. I particularly appreciate the addition of

organelle markers to facilitate determining MSC protein abundance changes over total organelle specific protein abundances not associated with MCSs.

- We thank the reviewer for dedicating additional time to the thorough reading and feedback on our manuscript, which has improved its interpretation and communication to a broad readership.

Minor comments

1. In the methods section, there is now an additional paragraph 'Proteomic comparisons of MCS-PRM data'. There are a few additions that would make this section more understandable to the reader.

a. How many markers are used– to save the reader trawling the supplementary tables

b. An explanation to the uninitiated what is meant by enhanced and supported scores from HPA

c. If DDA data was used from previous studies to the one described here, the authors should clearly state that the conditions used in these additional studies were the same as used here. If this was not the case, the authors should state the justification for their incorporation here.

- We have incorporated the reviewer's suggestions into the Methods section of the manuscript, clarifying the experimental and analytical foundations of the localization and functional comparisons for the MCS-PRM datasets for each infection. These changes can be found on page 45, lines 1033-1036, 1042-1044, and 1046-1052 in response to point a; lines 1040-1042 in response to point b, and lines 1058-1059 in response to point c. Of note, in accordance with suggestions from Reviewer 1, we further expanded the functional comparison, which now includes seven additional gene ontology terms, covering every major MCS function included in our MCS-PRM assay.

d. I'm afraid I'm a bit confused by figure S3A. I would clearly like to see the changes in abundance for the MCS protein with hpi/ virus and the organelle markers plotted separately, so it is clear if there are any organelle wide abundance changes upon viral infection. If I understand correctly - it is not very clearly started in the methods section – for each protein and each virus the PRM data are displayed in the left of the panel for each virus in S3A as PRM data and then PRM corrected for 'organelle median'. The response of each organelle marker should be shown in a heatmap too and some indications of the spread of data around the median. At the moment all I can see in table S7 is a pre-calculated organelle median – the same is true for the functional median.

- We agree with the reviewer that a graphical depiction of virus-induced changes to the abundance of organelle marker proteins would better illustrate the comparisons performed in Figure S3 and Table S7. To address this, we have now re-formatted Figure S3, dividing it into three figures: Figures S3 and S4 for localization and functional comparisons between organelle proteome and MCS changes, respectively, and Figure S5 for the median-corrected heatmaps. To Figures S3 and S4, we added panels B-C that shows the median and range of marker protein abundances during each infection (DDA-MS data from whole-proteome analysis), plotted with the median and range abundance changes of MCS proteins assigned to each category (MCS-PRM data from targeted MS analysis). We also have expanded the content in Table S7, which has the individual abundance ratios for each protein per timepoint per infection clearly listed (with gene names and accession numbers) next to the comparison of MCS-PRM to DDA-MS median abundance changes.

e. The data are qualitatively displayed, in so much there is no attempt to compare and results and in some cases there are some discrepancies between the two methods of calculating abundance change of MCS proteins along the infection timecourse. A statement that the trends are largely the same, but in some cases there are notable discrepancies for example, ORF8 Infl. A.

- We acknowledge that we did not elaborate as much as we could have on the proteomic comparisons in the Results section, and this was driven by our effort to streamline the manuscript to fit the journal length requirements. We have followed the reviewer's recommendation and have now edited the results section to include a more detailed description of the trends observed for MCS proteins compared to organelle and functional proteomes (pages 8-9). In addition, we have added a revised heatmap as a new Figure S5, which depicts protein abundances prior to and after comparisons.

The authors have gone some way to allay my concerns about overstating organelle localizations of MCS proteins when they could be present in more than one contact site and indeed, elsewhere in the cell.

I still think the authors should be more upfront about the limitations of their approach, however powerful, to manage the expectations of the researchers who use their PRM assays, along the lines of....'our approach gives indications of which MCSs are being modulated upon viral infections, but as some MCS proteins will be involved in multiple different tethering events, and may have alternate functions elsewhere in the cell, additional validity experiments such as microscopy and proximity ligation assays may be needed to support the data provided by PRM assays'.

- We agree that validation of any omic dataset, including targeted proteomics, is critical for biological interpretations, as we have thoroughly done in this study. As the reviewer suggested, we have edited the discussion to state useful validation metrics (page 26). Specifically, we indicated that "MCS proteins can be multi-localized, multi-functional, and contribute to multiple organelle interfaces (e.g., the master ER tethers VAP-A/B). Therefore, this assay can uncover the regulation of MCS proteins during a given biological process, helping to formulate functional hypotheses of multi-faceted organelle regulation. Additional validation using microscopy or proximity ligation assays, as shown in this study, may be needed to support the PRM data and gain specific insight into the localization of the tethering event." In addition, our introduction of the mitochondria and peroxisome follow-up studies include clear references to the hypothesis-generating purpose of MCS-PRM and subsequent need for validation. For example: "Using PLA, we tested whether increased ACBD5 and VAP-B abundances reflect enhanced organelle tethering..." (page 22, line 423) and "Our MCS-PRM findings enabled us to formulate hypotheses of how MCSs control the progression through different stages of the virus replication cycle.... To verify that the identified changes represent meaningful alterations in the extent of organelle interactions, we turned to IF microscopy..." (page 11, lines 264-265).

Reviewer #3 (Remarks to the Author):

Cook et al. explore the role of membrane contact sites (MCS) and define mitochondrial-endoplasmic reticulum encapsulation structures (MENCs) in the context of viral infection, using human herpesviruses, coronavirus, influenza A. This is an intriguing and well-carried out study with interesting and novel results that lay the foundation for new inquiries into MCSs and their role in viral infection. This study is powered by the innovation of mass spectrometry approaches to detect all MCSs with functional specificity and quantify changes in their components induced by viral infection. Major findings from this study include:

- HCMV induces a global increase in MCSs while other infections have a much less dramatic effect. By comparison, HSV-1 has little effect on MCSs and Influenza A and CoV-OC43 decreases MCSs.
- HCMV infection rewires mitochondrial-ER connections into encapsulated structures—MENCs
- MENCs are enriched for PTPIP51 and this contributes to HCMV replication, but loss of PTPIP51 does not

alter MENC structure.

- ER-mitochondrial tethering enhances STING-TBK1-IRF3 and HCMV reduces these contacts
- ER-peroxisome contacts are proviral

Issues for authors to address:

1. In the paragraph beginning on line 337, the authors state that they tested the anti-viral capacity (line 338) of increased ER-mito tethering. They show that tethering decreases HSV1, HCMV and InflA replication. The conclusion is that this result “suggests” that this is due to the increase in STING activation (line 345). This may be true, but the experiments performed provide only correlation and do not definitively define causation. Tethering could result in decreased dynamics of organelle trafficking or signaling or serve to stimulate other antiviral defenses not yet investigated. To firmly establish that the main consequence of tethering for infection is indeed increased STING signaling, can the defect in replication be fully rescued by STING knockdown in cells where tethering has been induced? Without this demonstration, tethering may restrict virus replication due to upregulation of STING, but other possibilities cannot be excluded. Then on line 524-526, it is stated: “we show that this anti-viral effect is due to enhanced activation of the STING-TBK1-IRF3 immune axis, preventing establishment of infection (Fig. 4).” But it really has not been shown to be due to only correlated with.

- We agree with the reviewer that more causative evidence for the mito-ER tether activation of STING would strengthen our manuscript. To address this, we measured virus titers in cells treated with known STING and TBK1 inhibitors: H-151 (Haag et al., 2018) and GSK8612 (Thomson et al., 2019), respectively. These small molecule inhibitors prevent phosphorylation of STING and/or TBK1, the downstream activation of IRF3, and the resulting production of interferons. Again, comparing virus production in mito-RFP-ER (tether) versus mito-BFP (control) cells, we find that STING and TBK1 inhibition results in partial rescue of virus production (new Figure S4J, with confirmation of expected drug treatment phenotypes in Figure S11G-H). Together with our other findings that mito-ER tethering causes STING relocalization, TBK1 and IRF3 phosphorylation, cytokine secretion, reduced virus gene expression, and delayed infection onset, this result demonstrates that the STING immune pathway is a contributor to the anti-viral effect of premature ER-mitochondria tethering. We have revised the text in the Results section to include this data (pages 20-21).

2. Figure 4 F and I both use heat maps. In F, dark purple indicates decreased abundance, whereas in panel I it indicates increased abundance. It would be helpful if all heat maps had the same colors indicating directionality especially since colors are coordinated throughout the manuscript.

- We thank the reviewer for catching this. We have updated the figure accordingly.

3. References are still need some curation so that the discussion of the results and interpretation in the greater context of the field is not so cursory. Two examples are given, but there are many places where the discussion could be deepened by providing more robust discussion to provide context.

-Example 1: Ref 50 is cited to support the statement that HCMV inhibits cell signaling by internalizing and degrading EGFR (line 188). Ref50 supports this statement to a point showing that HCMV reduces cell surface and total EGFR, but more detailed work has shown that EGFR is targeted for degradation and that it and downstream EGFR signaling is disrupted. PMID: 27218650 PMID: 30089695. Further, the MCSs described could be important to aberrant downstream signaling described for HCMV PMID: 32493823 PMID: 31725811,

activation of innate pathways PMID: 29021395, entry into latency PMID: 27974567, as well as trafficking virus capsids PMID: 32723814

-Example 2: Ref 69 and 70 are used to support the statement that HCMV inhibits STING (line 321). 69 is a paper on US9 and 70 is a review. There are a number of examples of HCMV gene products impacting cGAS/STING beyond US9 and that predate US9. PMID: 28132838 PMID: 29018427 PMID: 32466380. The regulation of cGAS/STING and its pathway in HCMV infection is very complicated and the cursory treatment of it in the discussion of the results does not do justice to how these findings contribute to the greater context of cGAS/STING. This is particularly true since STING has recently been shown to facilitate delivery of the capsid to the nucleus PMID: 34385328.

- We appreciate the reviewer's comment and detailed review of literature relevant to our manuscript. Many of the references listed here had been previously included but then deleted to trim the manuscript length to journal requirements. As the reviewer suggested, we have now added back many of the references listed above, as well as additional references for recent findings on MCSs, virus-driven organelle remodeling, and relevant functions. All of these new references are highlighted in the updated manuscript. In addition, we have expanded our results to include a more in-depth discussion of the contribution of ER-endosome contacts to the EGFR pathway during HCMV infection (pages 9-10), the complexity of STING during HCMV infection (pages 20-21), and place the exploration of ER-peroxisome contacts in the greater scheme of peroxisomal remodeling by human viruses (page 21, 25).

Reviewer #4 (Remarks to the Author):

The authors still failed to address my original concerns that there was no causative relationship provided between the virus titer and the STING immune signalling. The STING relocation was well demonstrated (e.g., in Fig. 4G), I agree, but it does not necessarily mean that the STING signalling suppressed the virus titer and increased the cytokine secretion.

As I suggested, the experiments should be performed with STING inhibitor or in STING-KO cells. Ideally, the experiments may also be performed with TBK1 inhibitor or in TBK1-KO cells.

- We agree with the reviewer that more causative evidence for the mito-ER tether activation of STING would strengthen our manuscript. To address this, we measured virus titers in cells treated with known STING and TBK1 inhibitors: H-151 (Haag et al., 2018) and GSK8612 (Thomson et al., 2019). These small molecule inhibitors prevent phosphorylation of STING and/or TBK1, the downstream activation of IRF3, and the resulting production of interferons. Again, comparing virus production in mito-RFP-ER (tether) versus mito-BFP (control) cells, we find that STING and TBK1 inhibition resulted in partial rescue of virus production (new Figure S4J, with confirmation of expected drug treatment phenotypes in Figure S11G-H). Together with our other findings that mito-ER tethering causes STING relocalization, TBK1 and IRF3 phosphorylation, cytokine secretion, reduced virus gene expression, and delayed infection onset, this result demonstrates that the STING immune pathway is a contributor to the anti-viral effect of premature ER-mitochondria tethering. We have revised the text in the Results section to include this data (pages 20-21).

REVIEWER COMMENTS

Reviewer #1 (Remarks to the Author):

While I appreciate the effort the authors have made to address my concerns, my assessment of this study remains the same. The hypothesis that all human viruses specifically regulate globally regulate membrane contact site (MCS) protein abundance is intriguing, but not strongly supported by the evidence. If the model were correct, then MCS protein abundance following infection should significantly change when compared to the abundance of proteins in similar functional categories. Fig. S4C shows this is only true of one of the 4 viruses characterized, HCMV. In addition, the study does not provide any indication of how even one virus specifically alters MCS protein abundance at all MCSs in a cell. The idea that all human viruses do is even more of a stretch. On the other hand, the evidence does show that some viruses could alter some MCSs to support propagation, an interesting new finding. But the study goes far beyond the data by claiming that human viruses generally regulate MCS protein abundance generally. These broader claims are not well supported by the evidence and seem implausible given the diverse strategies viruses employ to alter host cells.

Reviewer #3 (Remarks to the Author):

The revised manuscript has addressed my previous concerns. No issues remain.

Reviewer #4 (Remarks to the Author):

The authors adequately addressed my concerns. With STING and TBK1 inhibitors, the causative relationship between the virus titer and the STING immune signalling is suggested.

RESPONSE TO REVIEWERS

Reviewer #1 (Remarks to the Author):

While I appreciate the effort the authors have made to address my concerns, my assessment of this study remains the same. The hypothesis that all human viruses specifically regulate globally regulate membrane contact site (MCS) protein abundance is intriguing, but not strongly supported by the evidence. If the model were correct, then MCS protein abundance following infection should significantly change when compared to the abundance of proteins in similar functional categories. Fig. S4C shows this is only true of one of the 4 viruses characterized, HCMV. In addition, the study does not provide any indication of how even one virus specifically alters MCS protein abundance at all MCSs in a cell. The idea that all human viruses do is even more of a stretch. On the other hand, the evidence does show that some viruses could alter some MCSs to support propagation, an interesting new finding. But the study goes far beyond the data by claiming that human viruses generally regulate MCS protein abundance generally. These broader claims are not well supported by the evidence and seem implausible given the diverse strategies viruses employ to alter host cells.

>> We thank the reviewer for the time invested in reassessing our paper. We have addressed the remaining comments in several ways:

1. We agree with the reviewer that the most pronounced effects are seen following infection with HCMV, which we have stated in the manuscript. The changes observed for the other three viral infections tested are more subdued and more specific in nature (i.e., specific proteins, time points, or localizations). As the reviewer has indicated, this is expected given the unique features of the replication cycles of these viruses. We have followed the editor's and reviewer's suggestion by carefully rewording the claims throughout the manuscript to ensure that we avoid overstatements and to clarify the findings. We focused our descriptions on HCMV, and used the other infections primarily as comparison tools in our discussions.
2. In addition to the above mentioned text changes, we have removed our previous Figure 6A from the final model, which used to represent the overall changes in MCSs observed during the four viral infections tested. We are now focusing this figure on the two types of contacts that we functionally investigated, i.e., ER-mitochondria and ER-peroxisomes.
3. As a brief reminder, we did test the impact of selected MCSs of interest on the titers of the four viruses investigated. Specifically, we showed that ER-mitochondria tethering results in decreased production of HCMV, HSV-1, and influenza A. Additionally, we showed that ACBD5, the primary ER-peroxisome tether, impacted the titers of all four viruses tested (HCMV, HSV-1, influenza A, and OC43). Therefore, for these specific MCSs, we revised our text to enhance clarity, providing concrete information about their fold changes in each condition. The reviewer's comment led us to understand that the median protein abundance fold changes illustrated in the supplementary figures S3 and S4 miss the ability to observe the extent of protein-specific differences, which are better seen in Figures 1 and S5, and when considering the values in the Tables S3-S7. For example, for the specific subset of mitochondrial MCS proteins, we observed up to 5-fold increases during HCMV infection, while observing up to 2-fold, 3-fold, and 2-fold decreases in HSV-1, Infl. A, and HCoV-OC43 infections, respectively (page 10).
4. As stated above, we realized that perhaps the colors and graph lines did not properly convey the information about the level of changes in MCS protein abundances or their significance compared to proteins with similar localizations and functions. To clarify this, we have updated our supplementary figures S3 and S4 to include statistical analyses of the MCS-PRM and DDA-MS data comparisons. Specifically, have now added asterisks (*) to each graph indicating p-values for 1) one-way ANOVA with a Brown-Forsythe test for the cumulative temporal trends in MCS-PRM data after being compared to the DDA-MS data (shown on each graph); and 2) Dunnet's multiple comparisons test for each timepoint compared to Mock for the MCS-PRM data after comparison to the DDA-MS data (indicated below each graph for each timepoint).
5. Of additional consideration, the heatmaps in Figure S5A-B demonstrate that the abundance trends for a majority of MCS protein abundances do not change, or change very little, upon comparison to the DDA-MS data for all four infections (as indicated by an overall minimum or lack of difference between the

MCS-PRM data and the MCS-PRM data divided by DDA-MS data). The one exception would be HCMV, in which most trends become even further enhanced after comparison (indicated by the increased dark red coloring on the heatmaps). Overall, the minimal change to the protein abundance values after division by the DDA-MS dataset marker protein abundances indicates that that observed changes in MCS protein abundances are retained after the comparison to proteins of similar localization or function, even at the level of individual MCS proteins.

>> With regard to the reviewer's other comment, our study has focused on providing a method for studying MCSs, showing that this assay captures changes in organelle contacts across the space and time of infection. As follow-up experiments, we focused on characterizing ER-mediated contacts with peroxisomes, identifying the ER-peroxisome tether ACBD5 as being required for peroxisome remodeling during HCMV infection, and with mitochondria, investigating the new mitochondrial-ER encapsulation structure that we discovered, MENCs. Additionally, upon last revision and in accordance with this reviewer's concerns, we included a genetic investigation of the requirement for viral gene expression to control MCS protein abundances during HCMV infection (Figure S6). As a reminder, we used UV-irradiation to deactivate virus genomes as well as the clinical drug phosphonoformic acid (PFA) to inhibit late viral gene expression. Our findings indicate that HCMV infection relies on viral gene expression to alter MCS protein abundances, with UV-irradiation preventing nearly all abundance changes to MCS proteins, and PFA treatment acting in a protein-specific manner.

Reviewer #3 (Remarks to the Author):

The revised manuscript has addressed my previous concerns. No issues remain.

>> We thank the reviewer for their time and contributions to our manuscript.

Reviewer #4 (Remarks to the Author):

The authors adequately addressed my concerns. With STING and TBK1 inhibitors, the causative relationship between the virus titer and the STING immune signalling is suggested.

>> We thank the reviewer for their past suggestions to improve our manuscript.

RESPONSE TO REVIEWERS

Reviewer #1 (Remarks to the Author):

While I appreciate the effort the authors have made to address my concerns, my assessment of this study remains the same. The hypothesis that all human viruses specifically regulate globally regulate membrane contact site (MCS) protein abundance is intriguing, but not strongly supported by the evidence. If the model were correct, then MCS protein abundance following infection should significantly change when compared to the abundance of proteins in similar functional categories. Fig. S4C shows this is only true of one of the 4 viruses characterized, HCMV. In addition, the study does not provide any indication of how even one virus specifically alters MCS protein abundance at all MCSs in a cell. The idea that all human viruses do is even more of a stretch. On the other hand, the evidence does show that some viruses could alter some MCSs to support propagation, an interesting new finding. But the study goes far beyond the data by claiming that human viruses generally regulate MCS protein abundance generally. These broader claims are not well supported by the evidence and seem implausible given the diverse strategies viruses employ to alter host cells.

>> We thank the reviewer for the time invested in reassessing our paper. We have addressed the remaining comments in several ways:

1. We agree with the reviewer that the most pronounced effects are seen following infection with HCMV, which we have stated in the manuscript. The changes observed for the other three viral infections tested are more subdued and more specific in nature (i.e., specific proteins, time points, or localizations). As the reviewer has indicated, this is expected given the unique features of the replication cycles of these viruses. We have followed the editor's and reviewer's suggestion by carefully rewording the claims throughout the manuscript to ensure that we avoid overstatements and to clarify the findings. We focused our descriptions on HCMV, and used the other infections primarily as comparison tools in our discussions. Changes to the text included removing large sections from the Introduction (pages 3-4), Results (pages 8-11), and Discussion (page 28), and adding additional background information on HCMV infection in the Introduction (page 3) to enrich the understanding and interpretation of the HCMV-driven MCS changes. We also have re-titled the manuscript to reflect our specific findings during HCMV infection, and altered the figure titles and Result sub-section titles throughout the manuscript to focus on HCMV.
2. In addition to the above mentioned text changes, we have removed our previous Figure 6A from the final model, which used to represent the overall changes in MCSs observed during the four viral infections tested. We are now focusing this figure on the two types of contacts that we functionally investigated, i.e., ER-mitochondria and ER-peroxisomes.
3. As a brief reminder, we did test the impact of selected MCSs of interest on the titers of the four viruses investigated. Specifically, we showed that ER-mitochondria tethering results in decreased production of HCMV, HSV-1, and influenza A. Additionally, we showed that ACBD5, the primary ER-peroxisome tether, impacted the titers of all four viruses tested (HCMV, HSV-1, influenza A, and OC43). Therefore, for these specific MCSs, we revised our text to enhance clarity when the titering data is presented (pages 21 & 25), providing concrete information about their MCS protein fold changes in each condition. These few specific fold changes mentioned in the manuscript are values obtained after normalization/comparison to functional and localization proteome markers, as requested by this reviewer. We have removed claims that over-stated a correlative function for these MCSs in each infection.
4. The reviewer's comment led us to understand that the median protein abundance fold changes illustrated in the supplementary figures S3 and S4 miss the ability to observe the extent of protein-specific differences, which are better seen in Figures 1 and S5, and when considering the values in the Tables S3-S7. For example, for the specific subset of mitochondrial MCS proteins, we observed up to 5-fold increases during HCMV infection, while observing up to 2-fold, 3-fold, and 2-fold decreases in HSV-1, Infl. A, and HCoV-OC43 infections, respectively. As stated above, we realized that perhaps the colors and graph lines did not properly convey the information about the level of changes in MCS protein abundances or their significance compared to proteins with similar localizations and functions.

To clarify this, we have updated our supplementary figures S3 and S4 to include statistical analyses of the MCS-PRM and DDA-MS data comparisons. Specifically, we have now added asterisks (*) to each graph indicating p-values for 1) one-way ANOVA with a Brown-Forsythe test for the cumulative temporal trends in MCS-PRM data after being compared to the DDA-MS data (shown on each graph); and 2) Dunnett's multiple comparisons test for each timepoint compared to Mock for the MCS-PRM data after comparison to the DDA-MS data (indicated below each graph for each timepoint).

5. Of additional consideration, the heatmaps in Figure S5A-B demonstrate that the abundance trends for a majority of MCS protein abundances do not change, or change very little, upon comparison to the DDA-MS data for all four infections (as indicated by an overall minimum or lack of difference between the MCS-PRM data and the MCS-PRM data divided by DDA-MS data). The one exception would be HCMV, in which most trends become even further enhanced after comparison (indicated by the increased dark red coloring on the heatmaps). Overall, the minimal change to the protein abundance values after division by the DDA-MS dataset marker protein abundances indicates that the observed changes in MCS protein abundances are retained after the comparison to proteins of similar localization or function, even at the level of individual MCS proteins. However, we appreciate the reviewer's observation that many of these changes are slight for Influenza, HSV-1, and HCoV-OC43 infections, especially in comparison to the significant upregulation observed for nearly all MCS proteins during HCMV infection. Therefore, as stated above, we have removed nearly all discussion of the other infection datasets from our Results and Discussion sections, instead focusing on HCMV infection and only using the other infections as comparisons in the context of HCMV-driven changes.

>> With regard to the reviewer's other comment, our study has focused on providing a method for studying MCSs, showing that this assay captures changes in organelle contacts across the space and time of infection. As follow-up experiments, we focused on characterizing ER-mediated contacts with peroxisomes, identifying the ER-peroxisome tether ACBD5 as being required for peroxisome remodeling during HCMV infection, and with mitochondria, investigating the new mitochondrial-ER encapsulation structure that we discovered, MENCs. Additionally, upon last revision and in accordance with this reviewer's concerns, we included a genetic investigation of the requirement for viral gene expression to control MCS protein abundances during HCMV infection (Figure S6). As a reminder, we used UV-irradiation to deactivate virus genomes as well as the clinical drug phosphonoformic acid (PFA) to inhibit late viral gene expression. Our findings indicate that HCMV infection relies on viral gene expression to alter MCS protein abundances, with UV-irradiation preventing nearly all abundance changes to MCS proteins, and PFA treatment acting in a protein-specific manner.

Reviewer #3 (Remarks to the Author):

The revised manuscript has addressed my previous concerns. No issues remain.

>> We thank the reviewer for their time and contributions to our manuscript.

Reviewer #4 (Remarks to the Author):

The authors adequately addressed my concerns. With STING and TBK1 inhibitors, the causative relationship between the virus titer and the STING immune signalling is suggested.

>> We thank the reviewer for their past suggestions to improve our manuscript.